# Role of International Trade Competitive Advantage and Corporate Governance Quality in Predicting Equity Returns: Static and Conditional Model Proposals for an Emerging Market

Erol Muzir [1,*], Cevdet Kizil [1] and Burak Ceylan [2]

1   Department of Management, Faculty of Political Science, Istanbul Medeniyet University, Istanbul 34000, Turkey; cevdet.kizil@medeniyet.edu.tr
2   School of Graduate Studies, Istanbul Medeniyet University, Istanbul 34000, Turkey; ceylann.burakk@gmail.com
*   Correspondence: erol.muzir@medeniyet.edu.tr

**Abstract:** This paper aims to develop some static and conditional (dynamic) models to predict portfolio returns in the Borsa Istanbul (BIST) that are calibrated to combine the capital asset-pricing model (CAPM) and corporate governance quality. In our conditional model proposals, both the traditional CAPM (beta) coefficient and model constant are allowed to vary on a binary basis with any degradation or improvement in the country's international trade competitiveness, and meanwhile a new variable is added to the models to represent the portfolio's sensitivity to excess returns on the governance portfolio (BIST Governance) over the market. Some robust and Bayesian linear models have been derived using the monthly capital gains between December 2009 and December 2019 of four leading index portfolios. A crude measure is then introduced that we think can be used in assessing governance quality of portfolios. This is called governance quality score (GQS). Our robust regression findings suggest both superiority of conditional models assuming varying beta coefficients over static model proposals and significant impact of corporate governance quality on portfolio returns. The Bayesian model proposals, however, exhibited robust findings that favor the static model with fixed beta estimates and were lacking in supporting significance of corporate governance quality.

**Keywords:** static and conditional asset-pricing models; corporate governance quality; international trade capability; robust linear regression; Bayesian regression; Borsa Istanbul

**JEL Classification:** G11; G12; G17

## 1. Introduction

Modern portfolio theory explains how risk-averse investors select portfolios that are supposed to help them maximize expected returns in tune with their individual acceptable levels of portfolio risk. In this context, the capital markets theory attempts to provide a generic theoretical ground to reveal the relationship between expected return and investment risk. An optimal portfolio is defined and referred to as a particular portfolio providing a satisfactory expected return, given a certain level of risk (Markowitz 1952). It is assumed that all unsystematic or idiosyncratic risks can be eliminated or minimized through sufficient portfolio diversification, which leaves portfolios with systematic or undiversifiable risks only. Cognizant of this tenet, an investor should predict return and the systematic portion of risk by employing well-functioning tools or models. In the existing literature are many theories that try to serve this purpose, based on one or more (risk) factors as the determinants of expected return. These theories are determined to unveil functional relationships between returns and a presumed set of risk factors in such a way

that return sensitivities to these macro factors are estimated and then used in linear or nonlinear forms.

Regarding the available models of capital asset pricing, two fundamental approaches loom large in pricing stocks along with their various extensions in the academy and practice: the capital asset-pricing model (CAPM) and the arbitrage pricing theory (APT). These equilibrium-based theories are intended to supply a functional form of the relationship between risk and return and claim that higher stock returns are associated with higher levels of systematic risk.

The relevant literature is made up of a bulk of studies and model proposals focusing on security pricing, some of which can be viewed as efforts to develop special CAPM extensions, but others deal with constructing better multifactor models following the methodologies that the APT postulates. Among the most remarkable multifactor model proposals are the three-factor and five-factor asset-pricing models developed and empirically tested by Fama and French (2015, 1995), and the four-factor model of Carhart (1997). We can also mention some studies aimed at comparing the performances of alternative models with one another. The empirical findings and conclusions of the related studies carried out so far do not lead to a concrete convergence about which techniques and factors best serve for asset-pricing models.

In this context, this paper aims to develop some portfolio-based multifactor model proposals for the Borsa Istanbul (BIST) by adding a unique variable to the traditional form of CAPM in order to investigate the impact of corporate governance quality on systematic risk level and returns. This will allow the CAPM beta to vary conditionally with respect to changes in the real effective exchange rate index of Turkey, as an indicator of its international trade competitive power or advantage. Several static and conditional (semi-dynamic) models have been developed through both the frequentist and Bayesian approaches. Besides this, we introduce a plain measure of governance quality that can be used to compare the governance quality levels (or governance risk levels) of different stocks or portfolios.

All these aforementioned aspects of our study can be regarded as its major contributions to the existing literature, but the paper does not have the direct purpose of making a performance comparison between the CAPM and the APT. It debates whether it is feasible to simulate an APT model through a specific extension of CAPM that integrates a conventional version of the CAPM with the effects of competitive edge in foreign trade and governance quality on stock returns by deriving specific model proposals with a new predictor embodying the impact of corporate governance excellence and the market betas conditioned on changes in the real effective exchange rate index.

The manuscript is composed of six main sections and starts with two short sections that build brief conceptual and literature frameworks related to capital asset-pricing with emphasis on the CAPM and APT methodologies and models. The third section provides an overview of our empirical research regarding the data and model versions it involve, together with a concise introduction about the statistical approaches and techniques employed. Then follows another section that presents the empirical findings of the research with a straightforward proposal of governance quality score grounded on the model results just after a succinct account of the methodological issues. The fifth section is devoted to a broad compilation of the empirical results obtained, and the final section includes the concluding remarks and recommendations.

## 2. Conceptual Framework

Stock return predictions are of great importance in ascertaining corporate financial policies and investment decisions in that they are used as a proxy for the cost of equity in calculating the weighted average cost of capital as the discount rate to be employed in either firm valuation or evaluation of investment projects. The extent to which stock returns are accurately assessed is one of the foremost concerns of a financial manager (Berk and DeMarzo 2011). Thus, such financial asset-pricing models as the CAPM and APT are

being frequently utilized for this purpose, but since the numbers produced by such models are only an estimate or expected value, they are likely to have some deviations from the expected, which means those estimations may be speculative in some circumstances. For this reason, the analyst must carefully ensure that the models used meet all the theoretical and technical assumptions that they are based on. In this context, there may exist a concrete need to apply more robust approaches and techniques (Penman 2004).

The asset-pricing models assuming market equilibrium merely rely on the postulate that portfolio diversification helps elude unsystematic risks and as a result, investors have to deal only with how to manage systematic risks that they are exposed to. As is known, the CAPM as one of the paramount asset-pricing theories suggests the beta coefficient representing a stock's return sensitivity to changes in market returns to be the sole measure for return premium due to systematic risks. The theoretical CAPM relies on such basic economic principles as the law of one price and assumes the assets with identical levels of systematic risk to have the same expected return (Sharpe 1964). It describes expected asset return as a linear function of return on the market portfolio in excess of risk-free rate of return, so consider this excess return to be the single factor that represents systematic risk exposure. Furthermore, the coefficient associated with the market's excess return is called beta ($\beta_i$) and shows the sensitivity of asset returns to the market as a measure of systematic risk level. The simplest (ex-ante) form of the CAPM is given in the equation below, where $E(R_i)$ refers to expected rate of return on the asset $i$ while $E(R_m)$ and $R_f$ stand for expected return on the market portfolio and the risk-free rate of return, respectively.

$$E(R_i) = R_f + \beta_i \left[ E(R_m) - R_f \right] \tag{1}$$

To convert the ex-ante CAPM given above into its ex-post form. we should substitute Equation (1) in the empirical form of the CAPM depicted in Equation (2) where $R_{mt}$ and $R_{it}$ are referred to as actual rates of return on the market portfolio and the asset $i$ at time $t$. correspondingly, while $\varepsilon$ denotes error term. The resulting form (Equation (3) is the ex-post form of the CAPM.

$$R_{it} = E(R_{it}) + \beta_i[R_{mt} - E(R_{mt})] + \varepsilon_{it} \tag{2}$$

$$R_{it} - R_{ft} = \beta_i \left[ R_{mt} - R_{ft} \right] + \varepsilon_{it} \tag{3}$$

Despite the early empirical findings suggesting a significant linear and positive relationship between $\beta$ coefficients and realized returns, the restrictive assumptions of the CAPM (Fabozzi et al. 2010) have been frequently criticized in terms of their validity and underlie the rationale for oppositions to its accuracy. Therefore, there existed some remedial attempts to relax those unrealistic assumptions in order to overcome the major drawbacks they cause. Moreover, it has been strongly argued and questioned for lacking to capture the potential effects upon stock returns of important macro changes. The opponents also state that excess return on the market portfolio is the only factor used as risk premium, but more factors ought to be added to the model to improve prediction performance so as to cope with anomalies that the theory often fails to explain, arising from calendar and differences in firm size, market value of equity, and so forth (Elmiger 2019; Cadsbay 1992). This argument paved the way to exploring new multifactor extensions of the CAPM and the emergence of new asset-pricing theories.

The APT as an important alternative to the CAPM is based on the arbitrage argument that any deviations from theoretical expected returns suddenly disappear via the arbitrage mechanism and the market instantaneously reaches its equilibrium. The theory also suggests that expected returns can be determined by a variety of risk factors, maybe including excess return on the market portfolio (Ross 1976). As can be inferred, determination of true risk factors effective on asset returns is very critical and important to model success. With a set of $k$ assessed factors, the APT's typical functional form can be written as in Equation (4).

$$E(r_i) = \beta_{i1} E(r_{F1}) + \beta_{i2} E(r_{F2}) + \ldots + \beta_{ik} E(r_{Fk}) \tag{4}$$

where:

- $r_i$: return on the asset $i$ in excess of the risk-free rate of return;
- $\beta_{ij}$: measure for sensitivity of the asset $i$ returns to the $j$th systematic risk factor returns on the factor portfolio. with unit sensitivity to the $j$th risk factor and zero sensitivities to other factors);
- $r_{Fj}$: excess return of the factor portfolio over the risk-free rate.

The APT has relatively a lower number of restrictive assumptions and allows for the inclusion of macro factors other than the market's excess return in modeling. The theory provides a relatively flexible methodology to identify relevant factors effective on stock returns through such eligible statistical techniques as principal component analysis and factor analysis. It also consents to employing predetermined factors as predictors with no need to explore them via proper statistical and econometric methods.

In the face of a crowd of empirical findings impeaching validity of the CAPM, it seems marvelous that the CAPM-based models still continue to be used and researched. Nevertheless, the APT is said to be more valid and reliable as compared to the CAPM because the former builds upon no specific assumption about return distributions and affords the option to include more than one factor as predictor. Among its favorable features are also its dynamic methodology and that it does not require a hypothetical market portfolio consisting of all assets (Ross 1976).

Systematic risk factors can be determined by constructing variance–covariance matrices for asset returns and then calculating factor loadings. As an alternative to this approach, it is also possible to explore effective risk factors by using macroeconomic variables and their variations in calculating the factor loadings (Conner and Korajczyk 1993). Another option for setting up an APT model is to directly insert the macro variables into models that are known or believed to have outstanding influence on stock returns according to both the postulates of current theories and the findings of past research. The early studies concentrating on multifactor pricing models and the APT made use of such micro variables as return variation, firm size, and lagged returns in addition to the macroeconomic variables so as to test the viability of the APT and reported robust evidence suggesting insignificance of these additional variables as a proof for the validity of the APT (Roll and Ross 1980; Chen 1983). However, some exceptional research inferred the importance of firm size and other unsystematic risk factors in explaining stock returns, thereby challenging the profound insight that the APT models are appropriate and practicable (Reinganum 1981).

As a result, in the light of conceptual framework and our preceding discussions, it seems obviously beneficial to employ variables other than market risk premium as predictors in modeling stock returns through more sophisticated computation techniques, which will allow us to relax standard distribution assumptions.

### 3. Literature Review

The earlier studies carried out to test the CAPM exhibited some evidence supporting its accuracy and validity (Friend and Blume 1970; Black et al. 1972; Fama and MacBeth 1973). However, the subsequent research built up a framework challenging and disputing its soundness. The research referring to the validity of the CAPM drew attention to both the econometric problems experienced in model derivation and the fact that the theoretical assumptions were not satisfied in most cases. Moreover, it was stated that the CAPM becomes inaccurate and incompetent especially when error terms are not normally distributed and there exist significant correlations between them (Dimson 1979; Gibbons 1982). In a very recent research conducted by Ali and Badhani (2020) to empirically test the CAPM in the Indian equity market for investigating the presence of low-risk or low-beta anomalies, 650 traded stocks for the period covering 189 months starting from July 2002 and ending in March 2018 were studied by utilizing the Fama–MacBeth procedure. The research provided robust results through controlling outliers and correcting bias in standard errors. Thus, the study confirmed the existence of low-beta anomaly in India.

Furthermore, a nonlinear correlation was detected between the CAPM beta and expected returns.

Resulting further efforts to relax the assumptions about linearity; the presence of a risk-free asset and absence of non-traded assets; market imperfections; and frictions such as changing investor behaviors, non-negligible taxes, and transaction costs have led to the rollout of many CAPM variants and other multifactor models (Lintner 1969; Brennan 1970; Black 1972; Merton 1973). In the relevant literature are also some exceptional studies (Jaganathan and Wang 1996; Scruggs 1998; Ferson and Harvey 1999) suggesting that the CAPM beta values might vary in time because of changes in the macro environment conditions, which favor conditional models against the static ones. Some of the research addressed whether it could improve model performance and validity to include additional variables other than the beta coefficient, such as firm size, the market-to-book ratio and the squared beta, and provided sufficient evidence that inclusion of such new variables could positively affect prediction performance and model validity (Kothari et al. 1995; Fama and French 2002). Fama and French (2015, 1995) proposed the five-factor and three-factor models in which they used firm size, market-to-book ratio, momentum effect, profitability and investment dimensions as the predictors. Also, Carhart (1997) developed another model including monthly premium on winners minus losers in addition to three variables used by Fama and French in 1993.

The APT-related studies deal mainly with identification of significant macro variables taking roles in estimating systematic risk exposure. The relevant literature embodies lots of empirical findings shedding light on what these macro variables should be. To exemplify, Chen et al. (1986) stated that such macro variables as the industrial production index, default risk, yield curve shifts and unexpected inflation are among the systematic risk factors effective on stock returns. Some of the studies (Altay 2005; Dhankar and Singh 2005; Hu 2007; Sun and Zhang 2001; Muzır et al. 2010) aimed at comparing the performance of the CAPM and APT models with one another present empirical findings and conclusions on the reasons why the APT, in most cases, could outperform the CAPM, emphasizing its fewer assumptions and flexible methodology.

Nowadays, corporate governance as a contemporary management philosophy intended to reach the best fit between managerial activities and shareholder rights is densely associated with firms' market value. Recent studies point out a causality relationship among corporate governance, financial performance and stock returns and relate this relationship with market efficiency. Some of those studies (Black et al. 2006; Baker et al. 2009) equipped empirical evidence affirming significant positive connection between governance quality and return performance while some others provided somewhat contradictory findings. For instance, Ararat et al. (2017) in their research on the Borsa Istanbul found a significant strong correlation, but no causality relationship between governance quality and firm value. On the other hand, Bebchuck and Cohen (2005) and Hamza and Mselmi (2017) claimed that the relationship between governance and value may be negative because of high pressure on managers and increasing independence of the board of directors. In an exceptional research done by Diavatopoulos and Fodor (2016), no apparent and significant association between governance quality and stock returns could be identified.

The relationship between quality of government institutions and performance of global stock markets was also analyzed using international asset-pricing models in another study carried out by Hooper et al. (2009). A positive relationship was detected between quality of institutional environment and stock market performance. Average monthly stock index excess returns and the Sharpe ratio were utilized for performance measures. Additionally, the quality of governance was found to be negatively tied to stock market total risk and idiosyncratic risk. Moreover, it was detected that there was a relationship between stable institutions and reduced fluctuations in equity returns. Thus, countries armed with improved governance systems had stock markets with higher returns on equity and lower risk levels. The research also concluded that one of the leading necessities

of financial market development was the development of organizations that govern the exchange process.

As mentioned before, the CAPM and APT are indicated as the most common tools for predicting the opportunity cost of capital, which is one of the leading financial challenges for managers. In this context, the validity of the CAPM and APT to predict the cost of equity capital was discussed by Young and Saadi (2011). Based on the research, CAPM was still found to be the most effective and dominant tool for predicting the cost of equity.

Corporate governance and cost of equity relationship was investigated also by Teti et al. (2016) in a specific research that was run on Latin American companies. The degree of corporate governance systems' impact on cost of equity capital from the perspective of Latin American companies was analyzed. The research employed a corporate governance index and benefited from corporate governance quality-determining factors. Based on the results of this study, there was a negative relationship between corporate governance quality and the cost of equity. However, disclosure was the most eye-catching factor affecting the cost of equity. The research also emphasized that investors pay a higher premium to invest in companies with effective corporate governance standards in emerging and developing markets, compared to the developed markets.

According to Gupta et al. (2018), when the impact of corporate governance on firm value is analyzed, the distinction of external and internal governance is important. Based on their research, the effect of external governance on firm value is more apparent than that of internal governance. They used a sample consisting 7380 firm years gathered from 22 developed countries and found that the firm-level corporate governance factors and characters impact the cost of equity capital primarily in the Common Law countries, which reflect high levels of financial development.

Ok and Kim (2019) analyzed the effect of corporate social responsibility performance on the cost of equity in Korea. It was found that corporate governance affected the cost of equity. Besides, corporate governance was proved to increase firm value in Korea. Moreover, the result of research was robust in controlling for systematic risk, size, leverage ratio and the number of analysts.

The impact of firm corporate governance on lower stock-return volatility was investigated by Lee et al. (2019) as well. Panel data belonging to 1252 public listed firms in Asia covering 11 countries for 15 years were utilized in the research. The findings showed that corporate governance had a stabilizing impact on lower stock-return volatility. It was also found that better corporate governance would only decrease stock-return volatility for the companies with less foreign exposure.

There are some distinctive studies emphasizing the importance of corporate governance and its impact on the cost of capital. One of these studies stressed the fact that the significance of corporate governance became more evident following international financial crises, corruption and firm scandals since the 1980s. The mentioned study investigated the impact of corporate governance on the cost of capital by focusing on 76 manufacturing industry firms listed in Borsa Istanbul (BIST) between 2008 and 2017. The generalized method of moments (GMM) estimator developed by Arellano and Bover (1995) was used in the research. It was found that specific variables related to corporate governance had an impact on the cost of capital (Doğan and Acar 2020).

## 4. Empirical Research: Data, Models and Techniques

The empirical section of this paper is intended to develop several static and conditional models so as to predict portfolio returns and systematic risk level by extending the traditional CAPM in such a way that the CAPM beta coefficient is permitted to vary with changes in macroeconomic conditions and an additional unique variable specifically designated to represent the effect of corporate governance quality on returns is included. Our model proposals are a multifactor model in which the market risk premium and governance risk are integrated. The CAPM beta coefficient is estimated separately by dividing the entire period into two mutually exclusive sub-periods; the period with improving

national foreign trade competitive advantage and the period with degrading competitive advantage. In constructing model proposals, we prefer to use both the robust regression technique as a classical econometric approach and the Bayesian linear regression technique as a simulation method, which enables us to draw a comparison on model performances in terms of modeling approach. Independently, a new measure of governance quality is introduced as based on the empirical results of our model proposals that may be considered to be a simple indicator or comparison criterion.

To develop our model proposals in which capital gains (returns) are regressed on the market risk premium and corporate governance variables, we work with the monthly returns on four of the leading index portfolios listed in the Borsa Istanbul (BIST); the BIST 30, BIST Financials, BIST Industrials, and BIST Services which constitute a large fraction of all the listed firms in terms of size.

### 4.1. Sample Selection and Research Variables

The time period between December 2009 and December 2019 is taken as the sample period, because this period is assumed to be an interval when no extraordinary economic or financial crises occurred in Turkey. As a result, a data set consisting of 123 observations for each portfolio and each of the variables of interest have been gathered. Since portfolio capital gains are concerned, the effect of dividend gains on returns is ignored or assumed to be negligible.

After the monthly return rates were determined by using the simple logarithmic formula, the differences between the monthly returns and the risk-free rate of return ($r_f$) have been calculated for each portfolio and used as the dependent variable series (BIST 30, BIST Financials, BIST Industrials, and BIST Services). To approximate the risk-free rate, the 2-year TL treasury bond yields have been used as a benchmark. Then, they are regressed on two independent variables; MPREM calculated as the rate of return on the market portfolio (BIST All) minus the risk-free rate, and CGIOVERM defined as return on the Corporate Governance Index Portfolio (BIST Governance), consisting of all the listed firms with an overall governance quality score of 7 and over, in excess of the return on the market portfolio. For the purpose of capturing the effect of changes in Turkey's foreign trade competitive advantage on the portfolio's CAPM beta coefficient, two dummy variables, REERDUM1 and REERDUM2, are featured in the models. REERDUM1 is 1 in case of degraded competitive advantage, and 0 otherwise. Conversely. REERDUM2 takes on the value 1 in the event of improved competitiveness, but 0 otherwise. Degraded competitive advantage is concluded if there is any increase in the real effective exchange rate index value while improvement is assessed as no change, or any decrease in the index value (OECD 2020). By using these dummy variables, we plan to create dynamic model versions by allowing model constant and/or the beta coefficient to vary with respect to the time segments by degradation and improvement in the trade competitiveness (MPREM*REERDUM1 and MPREM*REERDUM2). This attempt is intended to help us question the empirical results of the previous researches (Topaloğlu and Karakozak 2018; Akçoraoğlu and Yurdakul 2002; Karatepe et al. 2002; Baillie and Cho 2016; Malliaropulos 1998; Wong 2017; Wong et al. 2018) which suggest remarkable variations of slope coefficients in the traditional CAPM with changes in economic circumstances, providing some evidence that such macroeconomic variables as the foreign exchange rate, foreign trade volume, interest rate, inflation, gold prices and money supply may have direct effects on stock returns and systematic risk levels.

In respect of the findings and results that our best model proposals provide, we aim to debate whether a static or conditional model performs better in predicting returns and systematic risk level by taking into account the conditional model constants (coefficients for REERDUM1 and REERDUM2) and MPREM coefficients (coefficients for the products of MPREM and REERDUM1 vs. REERDUM2) as well as their statistical significance. Moreover, through a comparative performance evaluation between the robust and Bayesian

models, it is also tested whether or not the Bayesian approach can surpass the frequentist (classical) approach.

*4.2. Model Versions and Econometric Techniques Employed*

The ex-post model versions that we study by using past data can be categorized as static and dynamic models. As mentioned before, model derivations have been accomplished through the robust and Bayesian linear regression methods.

4.2.1. Static and Dynamic Models

Equation (5) illustrates our static model version assuming no shift or value change in the model constant and slope coefficients, therefore their values remain fixed for the entire sample period. In the equation, $R_p$ refers to the portfolio's excess return over the risk-free rate while $\beta$ and $\theta$ stand for the coefficients of the market portfolio's excess return and excess return on the BIST Governance index portfolio over return on the market portfolio, respectively. Finally. $C$ is the model constant that is assumed to be the measure for the portfolio's return in excess of its expected return estimated by the model.

$$R_p = C + \beta(MPREM) + \theta(CGIOVERM) + \varepsilon \tag{5}$$

The second model version that allows the model constant to vary on a binary basis with any degradation or improvement in the country's competitive advantage is being expressed as a linear function in Equation (6). We call this version the 'conditional model with varying constant' where $C$ and $\alpha_1$ represents the model constants for the degradation and improvement periods, correspondingly.

$$R_p = C + \beta(MPREM) + \theta(CGIOVERM) + \alpha_1 \,(REERDUM1) + \varepsilon \tag{6}$$

Another one of our dynamic model versions is named 'conditional model with varying slopes', represented by Equation (7) where only the coefficient for the market's excess return is permitted to take on two different values with respect to whether a degradation or improvement has been experienced during the relevant month. In this context. $\beta_1$ denotes the market risk premium coefficient during degradation, but $\beta_2$ is referred to as the same coefficient in case of improved competitive advantage.

$$R_p = C + \beta_1(MPREM * REERDUM1) + \beta_2(MPREM * REERDUM2) + \theta(CGIOVERM) + \varepsilon \tag{7}$$

The most comprehensive dynamic model version casts value changes in both model constant and market risk premium slope coefficient. The functional form of this version is given in Equation (8).

$$R_p = C + \beta_1(MPREM * REERDUM1) + \beta_2(MPREM * REERDUM2) + \theta(CGIOVERM) + \alpha_1 \,(REERDUM1) + \varepsilon \tag{8}$$

4.2.2. Robust and Bayesian Regression Techniques: A Brief Introduction

The ordinary least squares (OLS) method has some drawbacks and pitfalls in challenging model accuracy and validity, especially whenever its technical assumptions related with form of relationships, predictors, variable distributions and error terms are not met. For instance, heteroscedasticity and autocorrelations among error terms may cause serious doubts about the reliability of coefficient tests. Hence, a better method is needed in deriving model functions that could yield more robust results than OLS would in cases where heteroscedasticity and autocorrelation occur. At this point, the robust linear regression method can be seen as a convenient modeling technique and is known to be able to provide equivalent or better performance even if the assumptions of OLS have all been satisfied. As contrary to OLS, it aims to minimize the weighted average or median of the error terms by utilizing such different approaches as the least median square, least trimmed square or weighted least square. There are some alternative algorithms such as M. S. or

MM-estimation to estimate model coefficients. Also, it offers some optional forms of target function. e.g., Huber or Bisquare, to minimize (Armutlulu and Yazıcı 2012).

To detect the best one from among the robust models, their Rw-square, deviance, and scale statistics as well as significance of coefficient estimates are all considered. The Rw-square statistic is interpreted in the same way as the R-square in OLS models. The higher the Rw-square and scale statistics are and the lower the deviance statistic is, the better the model. In assessing model validity, we take into account the Rn-square statistic and also pay attention to the extent to which the error terms are distributed normally.

The frequentist (classical econometric) approach strictly requires model variables and error terms to be normally distributed and reports static estimates (known values) for the variable coefficients. The available data drive the modeling process and it becomes very critical whether all the assumptions are met and model specifications are true. Interpretations and conclusions are built as based on the p-values calculated for the coefficient estimates.

The Bayesian approach as an alternative to the frequentist approach suggests that the true values of model parameters are not known and may change according to data sets to be used in modeling. Therefore, the approach does not provide a known value for any parameter. Available data contain some prior information about parameter distributions and the posterior (actual or true) distributions can be obtained or produced in light of both prior information the data contains and prior distribution assumptions about parameter distributions from existent theories, past experience, or empirical findings. To get an estimation for posterior distribution, a Markov Chain Monte Carlo (MCMC) simulation technique is chosen and used. At this point, there are several alternative algorithms such as the standard Metropolis–Hastings (MH), the adaptable random-walk MH, and the MH algorithm with Gibbs updates. It is also possible to increase the number of iterations or to block some parameters in order to boost model performance.

The simulation process results in such estimates as the mean, median, standard deviation, and standard error statistics as well as a credible interval for each coefficient or parameter. The specifications determined by the analyst regarding maximum number of iterations, type of simulation algorithm, and form of prior distribution for each parameter directly impact the success and overall efficiency of the modeling process, whereby results and conclusions are significantly altered.

The Bayesian modeling process begins with specifying prior and posterior distributions for the parameters. In the second stage, the MCMC simulation process is started and in the end, some reports are obtained and diagnosed to evaluate to what extent the process could generate ideal distributions for the parameters. In these reports are the efficiency scores and performance graphs pertaining to the parameters of interest. Furthermore, hypothesis tests are carried out on the credible intervals to decide whether generated posterior probabilities are identical to the ideal ones (STATACorp LLC 2019). Subsequent to ensuring realization of the ideal distributions, comparison of alternative models with each other in terms of prediction performance is necessary using two fundamental criteria; the posterior odds ratio (PO) and Bayes Factor (BF) (Berger 2000). The formulas used to calculate these measures are given in Equations (9) and (10). $P(y/M)$ refers to the marginal likelihood of the respective model, given the data $y$. $P(M/y)$ and $P(M)$ are respectively the model's posterior and prior probabilities. Relatively high PO and BF values indicate the superiority of the *ith model* to the *jth* model (Lee et al. 2007).

$$\text{PO}_{ij} = \frac{P(M_i/y)}{P(M_j/y)} = \frac{P(y/M_i)P(M_i)}{P(y/M_j)P(M_j)} \tag{9}$$

$$\text{BF}_{ij} = \frac{P(y/M_i)}{P(y/M_j)} \tag{10}$$

Since prior distribution assumptions have direct influence on model performance, it is recommended to start modeling with no prior assumption. The analyst should try various

types of prior distribution and aim to reach the best possible. To evaluate the performance of alternative models, the acceptance and efficiency rates can be taken into consideration. A higher acceptance rate is always required and preferred while any efficiency rate over 10% is found to be desirable, but efficiency rates between 1% and 10% may be acceptable, up to the analyst's judgment.

The Bayesian approach is preferable in cases that the classical approach techniques fail to model data or there is no standard econometric technique available for modeling. It is applicable to small data sets and facilitates the use of prior information and empirical findings in determining correct model specifications. However, the approach requires tedious and complex computations and a longer time to process data. Arbitrary choices made by the analyst are said to make the process relatively subjective and obtaining a sufficiently accurate model is probable only with enough experience and expertise (Bernardo and Smith 2000).

## 5. Methodology and Findings

### 5.1. Research Methodology

Before deriving our robust regression models, we included the descriptive statistics for the variables and then tested whether the variable distributions approximate a normal distribution through the Jarque–Bera and Doornik–Hansen tests. Additionally, stationarity of each series was argued using the augmented Dickey-Fuller (ADF) unit root test. Then, we questioned any multicollinearity problem among the predictor variables via the variance inflation factor (VIF) statistics. We did not need to conduct heteroscedasticity and autocorrelation tests because the robust regression technique can overcome the drawbacks due to these phenomena. Besides these tests, we preferred to employ the M-estimation algorithm in estimating model coefficients and the least trimmed square as the minimization approach. The bisquare option was chosen as the type of target function with the Huber Type I covariance matrix. For comparison between our robust model proposals, their Rw-square statistics were taken as a basis and we also paid attention to significance of the predictors (coefficients) in the model.

In deriving our Bayesian models, for each model version we developed four separate Bayesian linear models through MCMC simulations based on the standard MH algorithm with 10,000 iterations and 12,500 samples:

- Model Design 1: Model without any informative prior distribution assumption, but relying on the flat and Jeffrey's distributions;
- Model Design 2: Model with the standard normal distribution assumption;
- Model Design 3: Model with the multivariate normal distribution assumption proposed by Zellner and Revankar (1969);
- Model Design 4: Model with the multivariate normal distribution assumption with blocked variance and model constant.

Consequently, 16 models were derived for each portfolio. Their acceptance and overall efficiency rates were considered in selecting the best model design for each portfolio from among all these Bayesian models supposing any acceptance rate close to or over 40% and efficiency percentage over 1% to be reasonably satisfactory. In this manner, the model proposals with higher efficiency rates associated with the coefficients were deemed to be superior to others. Resultantly, we have come up with a candidate as the best model design for each model version as per portfolio. At the last step, we comparatively appraised the agreed best model designs to decide on the most prosperous model version. Then, we analyzed the efficiency scores and convergence graphs for the best-version model to judge its accuracy and validity. Through the credible interval hypothesis tests, the appropriateness of the posterior distribution proposed by the model was examined together with a standard coefficient estimate for every predictor.

Upon the determination of best robust and Bayesian model proposals for each portfolio, they were compared as to their coefficients of determination as well as other performance criteria such as the sum of squared errors (RSS), mean squared errors (MSE), Theil's

$R^2$, Amemiya's prediction criterion (PC), and Akaike information criterion (AIC) (Muzır and Çağlar 2009).

Based on the coefficient estimates (θ) and their p-values (available only in case of robust model) for *CGIOVERM*, we put forward an elementary overall measure of corporate governance quality for portfolios, called the governance quality score (GQS). To calculate this score, the typical sigmoid function depicted in Equation (11) was utilized where $\gamma = \theta/(1 + p\text{-value})$. As can be inferred from the equation, the score value is expected to vary between 0 and 1 and will increase as the coefficient estimate rises, but decrease with any decline in the estimate. Decaying significance of the coefficient simultaneously leads to a lower score while the score will get closer to 1 if the estimate becomes less reliable, i.e., with larger *p*-values. On the other hand, since Bayesian models do not confer p-values for coefficient estimates, we substituted the standard coefficient estimates (standardized θs) into the equation for $\gamma$.

$$GQS_i = \frac{e^\gamma}{e^\gamma + 1} = \frac{1}{1 + e^{-\gamma}} \tag{11}$$

This proposed measure may be a numerical criterion to consider in assessing changes in the governance quality of a portfolio and to make a basic comparison between separate portfolios in terms of governance quality.

*5.2. Empirical Findings*

Prior to modeling, we argued whether the data set at hand confirms the distribution requirements of robust regression by undertaking normal distribution and stationarity (unit root) tests. Table 1 presents the descriptive statistics and goodness-of-fit test results for the data. The findings suggest that all the variable distributions seem to approximate a normal distribution because the univariate Jarque–Bera and multivariate Doornik–Hansen test statistics are not significant at the 5% significance level. All of the variable distributions proved to be symmetric enough. These findings can be considered favorable for our robust regression practices.

**Table 1.** Descriptive statistics and distribution tests.

| Variable: | BIST 30 | BIST Financials | BIST Services | BIST Industrials | MPREM | CGIOVERM |
|---|---|---|---|---|---|---|
| Average | −0.003397 | −0.005064 | −0.001486 | 0.001585 | −0.002409 | −0.008644 |
| Median | −0.002391 | −0.001210 | 0.002382 | 0.006190 | −0.000367 | −0.008855 |
| Maximum | 0.134039 | 0.151315 | 0.123426 | 0.117553 | 0.115626 | 0.021447 |
| Minimum | −0.148899 | −0.193182 | −0.148394 | −0.182615 | −0.152541 | −0.045002 |
| Standard Deviation | 0.066399 | 0.074329 | 0.057695 | 0.057279 | 0.062008 | 0.013012 |
| Skewness | −0.050167 | −0.055367 | −0.441077 | −0.397933 | −0.171635 | 0.166896 |
| Kurtosis | 2.294102 | 2.285280 | 2.599989 | 2.993221 | 2.250969 | 2.712600 |
| Sum | −0.417840 | −0.622894 | −0.182746 | 0.194913 | −0.296333 | −1.063158 |
| Sum of Squares | 0.537875 | 0.674027 | 0.406109 | 0.400263 | 0.469083 | 0.020655 |
| Observations | 123 | 123 | 123 | 123 | 123 | 123 |
| **Univariate Normal Distribution Tests** | | | | | | |
| Jarque–Bera Statistic | 2.605 | 2.680 | 4.808 | 3.246 | 3.479 | 0.994 |
| Significance (*p*-value) | 0.2718 | 0.2617 | 0.0903 | 0.1972 | 0.1755 | 0.6082 |
| **Multivariate Normal Distribution Tests (Dependent Variable feat. MPREM and CGIOVERM)** | | | | | | |
| Doornik–Hansen Statistic | 5.798 | 12.319 | 5.214 | 6.237 | | |
| Significance (*p*-value) | 0.4462 | 0.0552 | 0.5167 | 0.3972 | | |

Source: Eviews output (author's compilation).

The univariate ADF unit root test results given in Table 2 provide robust evidence to conclude stationarity for the variable distributions. Considering their significant ADF statistics, all the variables seem to have no unit roots at level in every case including a

constant and/or a time trend, which convinces us that it has no trouble working with time series to model relationships in linear form.

**Table 2.** Augmented Dickey-Fuller unit root test statistics.

| Variable | Including Constant (at Level) | | Including Constant and Time Trend (at Level) | |
|---|---|---|---|---|
| | **ADF Statistic** | **Prob (*p*-Value)** | **ADF Statistic** | **Prob (*p*-Value)** |
| BIST 30 | −12.05041 | 0.000 | −12.02601 | 0.000 |
| BIST Financials | −12.10318 | 0.000 | −12.07932 | 0.000 |
| BIST Services | −12.83095 | 0.000 | −12.78004 | 0.000 |
| BIST Industrials | −11.10752 | 0.000 | −11.10160 | 0.000 |
| MPREM | −11.94444 | 0.000 | −11.79790 | 0.000 |
| CGIOVERM | −10.21594 | 0.000 | −10.94135 | 0.000 |

Source: Eviews output (author's compilation).

Table 3 depicts the VIF scores calculated to ascertain any multicollinearity problem between the predictors, which might challenge model accuracy. We are assured that there is no multicollinearity according to the table findings because the VIF scores are very low and much below 5.

**Table 3.** Variance inflation factors (VIF).

| Dependent Variable | Model Version | Independent Variable | VIF Statistics | | |
|---|---|---|---|---|---|
| | | | **Variance** | **Uncentered VIF** | **Centered VIF** |
| BIST30 | Static | MPREM | 0.000162 | 1.001781 | 1.000259 |
| | | CGIOVERM | 0.003677 | 1.445283 | 1.000259 |
| | Dynamic with Varying Slope | MPREM*REERDUM1 | 0.000364 | 1.071158 | 1.005423 |
| | | MPREM*REERDUM2 | 0.000296 | 1.087822 | 1.006221 |
| | | CGIOVERM | 0.003438 | 1.446268 | 1.000941 |
| BIST Financials | Static | MPREM | 0.000609 | 1.001781 | 1.000259 |
| | | CGIOVERM | 0.013832 | 1.445283 | 1.000259 |
| | Dynamic with Varying Slope | MPREM*REERDUM1 | 0.001459 | 1.071158 | 1.005423 |
| | | MPREM*REERDUM2 | 0.001185 | 1.087822 | 1.006221 |
| | | CGIOVERM | 0.013780 | 1.446268 | 1.000941 |
| BIST Industrials | Static | MPREM | 0.000715 | 1.001781 | 1.000259 |
| | | CGIOVERM | 0.016231 | 1.445283 | 1.000259 |
| | Dynamic with Varying Slope | MPREM*REERDUM1 | 0.001730 | 1.071158 | 1.005423 |
| | | MPREM*REERDUM2 | 0.001406 | 1.087822 | 1.006221 |
| | | CGIOVERM | 0.016343 | 1.446268 | 1.000941 |
| BIST Services | Static | MPREM | 0.001532 | 1.001781 | 1.000259 |
| | | CGIOVERM | 0.034792 | 1.445283 | 1.000259 |
| | Dynamic with Varying Slope | MPREM*REERDUM1 | 0.003654 | 1.071158 | 1.005423 |
| | | MPREM*REERDUM2 | 0.002969 | 1.087822 | 1.006221 |
| | | CGIOVERM | 0.034522 | 1.446268 | 1.000941 |

Source: Eviews output (author's compilation).

### 5.2.1. Robust Linear Regression Findings

Respecting their Rn-square statistics, our robust model proposals are all determined to be accurate at the 1% significance level. But for all of the portfolios, the conditional models with varying slopes outperform other version models if we consider their corresponding Rw-square statistics as well as significance of the predictors. (Table 4). This finding suggests that it would rather adjust the beta coefficient in accordance with the country's foreign trade competitiveness position.

**Table 4.** Comparison of robust regression models.

| Dependent Variable | Performance Measure | Robust Model Version | | | |
|---|---|---|---|---|---|
| | | Static Model | Conditional with Varying Constant | Conditional with Varying Slopes | Conditional with Varying Constant & Slopes |
| BIST 30 | Rn-Square Statistic | 8668.585 (0.0000) | 8557.604 (0.0000) | 8888.416 (0.0000) | 8810.190 (0.0000) |
| | Rw-Square | 0.9897 | 0.9896 | 0.9897 | 0.9897 |
| | Adjust Rw-Square | 0.9896 | 0.9896 | 0.9897 | 0.9897 |
| BIST Financials | Rn-Square Statistic | 2253.665 (0.0000) | 2232.792 (0.0000) | 2304.384 (0.0000) | 2279.748 (0.0000) |
| | Rw-Square | 0.9592 | 0.9591 | 0.9609 | 0.9609 |
| | Adjust Rw-Square | 0.9592 | 0.9591 | 0.9609 | 0.9609 |
| BIST Industrials | Rn-Square Statistic | 1020.097 (0.0000) | 1024.952 (0.0000) | 1005.239 (0.0000) | 1012.674 (0.0000) |
| | Rw-Square | 0.9135 | 0.9143 | 0.9138 | 0.9142 |
| | Adjust Rw-Square | 0.9135 | 0.9142 | 0.9138 | 0.9142 |
| BIST Services | Rn-Square Statistic | 404.819 (0.0000) | 403.537 (0.0000) | 410.466 (0.0000) | 409.691 (0.0000) |
| | Rw-Square | 0.8062 | 0.8081 | 0.8090 | 0.8119 |
| | Adjust Rw-Square | 0.8062 | 0.8081 | 0.8089 | 0.8119 |

Source: STATA output (author's compilation). The values in parentheses in the table represent *p*-values.

Table 5 summarizes the outputs of our conditional robust models with varying slopes to enable an integrated analysis. The coefficient estimates in all prove to be significant at 5%. The model constants can be said to be significant in the models except the model for BIST Services, BIST 30 and BIST Financials are two portfolios that generated an average rate of return below what the respective model predicted. On the contrary, BIST Industrials yielded a return, on average, above that predicted by the model. BIST Services generated no excess return over that predicted. BIST 30 and BIST Financials constitute two index portfolios whose sensitivities to market risk premium were estimated to increase for the sub-period with degraded competitiveness, but BIST Industrials and BIST Services deviate from the former two and appear to have lower beta coefficients for the same sub-period. Returns on BIST 30, BIST Financials and BIST Services are negatively influenced by excess returns on BIST Governance over the market while the BIST Industrials returns are in positive association with them. Finally, the error terms are found to be normally distributed at 5% for BIST Industrials and BIST Services, but we cannot conclude the same for BIST 30 and BIST Financials. The highest and lowest coefficients of determination are for BIST 30 and the BIST Services respectively.

**Table 5.** Robust regression models: summary and performance.

| Dependent Variable | Conditional Robust Model with Varying Slopes | | | |
|---|---|---|---|---|
| | BIST30 | BIST Financials | BIST Industrials | BIST Services |
| C | −0.002700 (0.0023) | −0.006651 (0.0004) | 0.012185 (0.0000) | −0.001898 (0.5513) |
| CGIOVERM | −0.128954 (0.0175) | −0.459450 (0.0001) | 0.957911 (0.0000) | −0.418115 (0.0321) |
| MPREM*REERDUM1 | 1.114347 (0.0000) | 1.203477 (0.0000) | 0.842487 (0.0000) | 0.756043 (0.0000) |
| MPREM*REERDUM2 | 1.037636 (0.0000) | 1.113536 (0.0000) | 0.849429 (0.0000) | 0.878022 (0.0000) |

| Dependent Variable | Conditional Robust Model with Varying Slopes | | | |
|---|---|---|---|---|
| | BIST30 | BIST Financials | BIST Industrials | BIST Services |
| **Performance Statistics** | | | | |
| Rn-Square Statistic | 8888.416 | 2304.384 | 1005.239 | 410.466 |
| | (0.0000) | (0.0000) | (0.0000) | (0.0000) |
| MSE | 0.008456 | 0.016924 | 0.018397 | 0.026692 |
| R-Square | 0.8119 | 0.8307 | 0.7645 | 0.6943 |
| Adjusted R-Square | 0.8072 | 0.8264 | 0.7588 | 0.6866 |
| Rw-Square | 0.9897 | 0.9609 | 0.9138 | 0.8090 |
| Adjust Rw-Square | 0.9897 | 0.9609 | 0.9138 | 0.8089 |
| Jarque-Bera | 6.762821 | 13.17274 | 0.271160 | 1.077918 |
| Statistic(Residuals) | (0.0339) | (0.0014) | (0.8732) | (0.5834) |

Source: STATA output (author's compilation). The values in parentheses represent *p*-values.

5.2.2. Bayesian Linear Regression Findings

Looking at the acceptance and efficiency rates calculated for our Bayesian model proposals given in Table 6 as per model version and design, we can deduce superiority of the model design with multivariate normal distribution assumption and blocking option (Model Design 4) over the other model versions, from their relatively high acceptance and efficiency scores. After selecting the best of each model version from among the available model designs, we examined the most successful model version in respect of the posterior odds ratios and Bayes factor statistics attributed to four alternative model versions, which are summarized in Table 7. As can be apparently understood from the findings, the static model version has greater Bayes factor (BF), maximum likelihood (ML), and P(M/y) values, but lower deviance information criterion (DIC) scores for all of the portfolios and hence surpasses the other versions. Our Bayesian results indicate no need for a conditional approach in computing the beta coefficient so that it is estimated to remain constant or reveal no significant change during the whole sample period.

The details regarding our static Bayesian models are presented in Table 8 including the mean, median, standardized value, standard deviation and error, credible interval, and efficiency rate for each model parameter.

The MPREM coefficient mean estimates in the above models show that all the portfolios except BIST Financials have a systematic risk level below that of the market portfolio, i.e., coefficient estimates smaller than 1. Since the credible intervals determined for the MPREM coefficient estimates do not include 0, this variable can be considered statistically significant. Besides, the CGIOVERM coefficient estimates suggest an inverse influential relationship between excess returns on BIST Governance over the market and returns on the BIST 30, BIST Financials, and BIST Services index portfolios. On the other hand, this relationship is found positive for the BIST Industrials portfolio. But we suspect insignificance of this predictor as its assumed credible intervals involve 0 in every case. The same can be claimed for the model constant, which provides some evidence to conclude none of the portfolios could generate a return, on average, over that predicted by the respective model. For as much as the efficiency rates calculated for the parameters are approximately 7% and over, all the static model proposals prove to be reasonably efficient.

**Table 6.** Bayesian models: performance evaluation.

| Model Version | Performance Crtierion | BIST 30 | | | | BIST Financials | | | | BIST Industrials | | | | BIST Services | | | |
|---|---|---|---|---|---|---|---|---|---|---|---|---|---|---|---|---|---|
| | | Model Design 1 | Model Design 2 | Model Design 3 | Model Design 4 | Model Design 1 | Model Design 2 | Model Design 3 | Model Design 4 | Model Design 1 | Model Design 2 | Model Design 3 | Model Design 4 | Model Design 1 | Model Design 2 | Model Design 3 | Model Design 4 |
| Static | Acceptance Rate | 0.1985 | 0.2250 | 0.2441 | 0.3950 | 0.2623 | 0.2725 | 0.2420 | 0.3555 | 0.2222 | 0.1423 | 0.1812 | 0.3647 | 0.2437 | 0.2005 | 0.2260 | 0.4134 |
| | Minimum Efficiency | 0.0013 | 0.0342 | 0.0527 | 0.0783 | 0.0258 | 0.0450 | 0.0540 | 0.0681 | 0.0454 | 0.0471 | 0.0435 | 0.0765 | 0.0443 | 0.0364 | 0.0239 | 0.0714 |
| | Average Efficiency | 0.0514 | 0.0589 | 0.0690 | 0.1177 | 0.0621 | 0.0644 | 0.0669 | 0.1211 | 0.0598 | 0.0582 | 0.0498 | 0.1202 | 0.0579 | 0.0640 | 0.0614 | 0.1272 |
| | Maximum Efficiency | 0.1128 | 0.0859 | 0.0893 | 0.2061 | 0.0858 | 0.0920 | 0.0797 | 0.2050 | 0.0789 | 0.0769 | 0.0589 | 0.1909 | 0.0699 | 0.0933 | 0.0791 | 0.2181 |
| Dynamic with Varying Constant | Acceptance Rate | 0.1908 | 0.2403 | 0.2510 | 0.3506 | 0.2061 | 0.2278 | 0.2286 | 0.3674 | 0.2021 | 0.1787 | 0.2111 | 0.3936 | 0.2133 | 0.2110 | 0.2274 | 0.3925 |
| | Minimum Efficiency | 0.0013 | 0.0412 | 0.0375 | 0.0450 | 0.0445 | 0.0378 | 0.0337 | 0.0350 | 0.0017 | 0.0318 | 0.0220 | 0.0464 | 0.0223 | 0.0463 | 0.0031 | 0.0411 |
| | Average Efficiency | 0.0172 | 0.0525 | 0.0565 | 0.0779 | 0.0538 | 0.0568 | 0.0495 | 0.0793 | 0.0266 | 0.0418 | 0.0444 | 0.0904 | 0.0534 | 0.0534 | 0.0289 | 0.0849 |
| | Maximum Efficiency | 0.0688 | 0.0728 | 0.0883 | 0.1882 | 0.6507 | 0.0711 | 0.0636 | 0.1774 | 0.0531 | 0.0537 | 0.0765 | 0.2048 | 0.0878 | 0.0647 | 0.0573 | 0.1956 |
| Dynamic with Varying Slope | Acceptance Rate | 0.2195 | 0.2698 | 0.1951 | 0.3771 | 0.1803 | 0.2486 | 0.1470 | 0.3453 | 0.2850 | 0.2259 | 0.2383 | 0.3596 | 0.2323 | 0.2271 | 0.3043 | 0.3670 |
| | Minimum Efficiency | 0.0025 | 0.0066 | 0.0317 | 0.0464 | 0.0018 | 0.0254 | 0.0025 | 0.0472 | 0.0039 | 0.0429 | 0.0337 | 0.0604 | 0.0039 | 0.0106 | 0.0231 | 0.0500 |
| | Average Efficiency | 0.0258 | 0.0186 | 0.0478 | 0.0968 | 0.0169 | 0.0440 | 0.0228 | 0.0830 | 0.0393 | 0.0509 | 0.0430 | 0.1013 | 0.0263 | 0.0494 | 0.0420 | 0.0821 |
| | Maximum Efficiency | 0.0753 | 0.0288 | 0.0656 | 0.2140 | 0.0597 | 0.0579 | 0.0532 | 0.1574 | 0.0711 | 0.0568 | 0.0732 | 0.1888 | 0.0511 | 0.0787 | 0.1055 | 0.1562 |
| Dynamic with Varying Constant and Varying Slope | Acceptance Rate | 0.2825 | 0.2741 | 0.2514 | 0.3911 | 0.2001 | 0.2182 | 0.2948 | 0.3962 | 0.2244 | 0.1781 | 0.1395 | 0.3825 | 0.2740 | 0.2541 | 0.3019 | 0.3755 |
| | Minimum Efficiency | 0.0104 | 0.0015 | 0.0071 | 0.0188 | 0.0139 | 0.0166 | 0.0243 | 0.0319 | 0.0195 | 0.0027 | 0.0213 | 0.0334 | 0.0142 | 0.0026 | 0.0131 | 0.0255 |
| | Average Efficiency | 0.0336 | 0.0245 | 0.0228 | 0.0680 | 0.0306 | 0.0324 | 0.0343 | 0.0714 | 0.0280 | 0.0068 | 0.0361 | 0.0668 | 0.0209 | 0.0154 | 0.0302 | 0.0615 |
| | Maximum Efficiency | 0.0513 | 0.1201 | 0.0325 | 0.1866 | 0.0471 | 0.0380 | 0.0453 | 0.1723 | 0.0400 | 0.0119 | 0.0546 | 0.1851 | 0.0359 | 0.0402 | 0.0491 | 0.1757 |

Source: STATA output (author's compilation).

**Table 7.** Bayesian models: posterior odds ratios and Bayes factors.

| Model Version | BIST 30 | | | |
| --- | --- | --- | --- | --- |
| | DIC | LOG(ML) | LOG(BF) | P(M/y) |
| Static Model | −104.6688 | −10.2670 | 0.0000 | 0.6438 |
| Dynamic Model with Varying Constant | −102.9783 | −11.5648 | −1.2979 | 0.1758 |
| Dynamic Model with Varying Slopes | −102.4693 | −11.5390 | −1.2720 | 0.1804 |
| Dynamic Model with Varying Constant and Slopes | −73.0189 | −26.3844 | −16.1175 | 0.0000 |

| Model Version | BIST Financials | | | |
| --- | --- | --- | --- | --- |
| | DIC | LOG(ML) | LOG(BF) | P(M/y) |
| Static Model | −138.2992 | 6.9186 | 0.0000 | 1.0000 |
| Dynamic Model with Varying Constant | −101.5432 | −11.8150 | −18.7336 | 0.0000 |
| Dynamic Model with Varying Slopes | −101.6660 | −11.7617 | −18.6803 | 0.0000 |
| Dynamic Model with Varying Constant and Slopes | −72.8041 | −26.7789 | −33.6975 | 0.0000 |

| Model Version | BIST Industrials | | | |
| --- | --- | --- | --- | --- |
| | DIC | LOG(ML) | LOG(BF) | P(M/y) |
| Static Model | −138.8032 | 7.7105 | 0.0000 | 1.0000 |
| Dynamic Model with Varying Constant | −101.1873 | −11.6099 | −18.7151 | 0.0000 |
| Dynamic Model with Varying Slopes | −102.5216 | −11.7487 | −18.8539 | 0.0000 |
| Dynamic Model with Varying Constant and Slopes | −72.9679 | −26.4737 | −33.5789 | 0.0000 |

| Model Version | BIST Services | | | |
| --- | --- | --- | --- | --- |
| | DIC | LOG(ML) | LOG(BF) | P(M/y) |
| Static Model | −137.2563 | 6.6372 | 0.0000 | 1.0000 |
| Dynamic Model with Varying Constant | −100.3344 | −11.9132 | −18.5504 | 0.0000 |
| Dynamic Model with Varying Slopes | −100.5585 | −11.9654 | −18.6026 | 0.0000 |
| Dynamic Model with Varying Constant and Slopes | −72.4734 | −26.9859 | −33.6231 | 0.0000 |

Source: STATA output (author's compilation).

**Table 8.** Static Bayesian models: summary.

| Variable/Parameter | BIST 30 Bayesian Model Statistics | | | | | | | | |
| --- | --- | --- | --- | --- | --- | --- | --- | --- | --- |
| | Mean | Standardized Value | Standard Deviation | Standard Error (MCSE) | Median | 95% Credible Interval Lower Limit | 95% Credible Interval Upper Limit | Effective Sample Size ESS | Efficiency Rate |
| MPREM | 0.9780260 | 0.9144418 | 0.3643173 | 0.012363 | 0.9736101 | 0.2973357 | 1.6986190 | 868.32 | 0.0868 |
| CGIOVERM | −0.0777509 | −0.0530367 | 1.7260920 | 0.061694 | −0.1295789 | −3.224778 | 3.326044 | 782.78 | 0.0783 |
| C | −0.0006968 | | 0.0263154 | 0.000834 | −0.0009780 | −0.0519224 | 0.0517887 | 996.20 | 0.0996 |
| Variance (Sigma2) | 0.0658121 | | 0.0083355 | 0.000184 | 0.0649997 | 0.0513910 | 0.0834123 | 2060.98 | 0.2061 |

| Variable/Parameter | BIST Financials Bayesian Model Statistics | | | | | | | | |
| --- | --- | --- | --- | --- | --- | --- | --- | --- | --- |
| | Mean | Standardized Value | Standard deviation | Standard Error (MCSE) | Median | 95% Credible Interval Lower Limit | 95% Credible Interval Upper Limit | Effective Sample Size ESS | Efficiency Rate |
| MPREM | 1.0776480 | 0.7186622 | 0.3051736 | 0.009648 | 1.0840250 | 0.4674241 | 1.6654280 | 1000.56 | 0.1001 |
| CGIOVERM | −0.3832704 | −0.4156208 | 1.4536070 | 0.055715 | −0.3539888 | −3.2738730 | 2.5253270 | 680.68 | 0.0681 |
| C | −0.0054632 | | 0.0229047 | 0.000687 | −0.0059224 | −0.0513845 | 0.0392210 | 1113.18 | 0.1113 |
| Variance (Sigma2) | 0.04998706 | | 0.0063380 | 0.000140 | 0.0493739 | 0.0385422 | 0.0633937 | 2050.46 | 0.2050 |

| Variable/Parameter | BIST Industrials Bayesian Model Statistics | | | | | | | | |
| --- | --- | --- | --- | --- | --- | --- | --- | --- | --- |
| | Mean | Standardized Value | Standard Deviation | Standard Error (MCSE) | Median | 95% Credible Interval Lower Limit | 95% Credible Interval Upper Limit | Effective Sample Size ESS | Efficiency Rate |
| MPREM | 0.7789489 | 0.8432594 | 0.30121330 | 0.009127 | 0.7611310 | 0.1935407 | 1.3790210 | 1089.27 | 0.1089 |
| CGIOVERM | 1.0271850 | 0.2333382 | 1.4447280 | 0.052227 | 1.0849470 | −1.8068540 | 3.9258310 | 765.22 | 0.0765 |
| C | 0.0128308 | | 0.0230956 | 0.000714 | 0.0128392 | −0.0338013 | 0.0585056 | 1046.26 | 0.1046 |
| Variance (Sigma2) | 0.0494855 | | 0.0064047 | 0.000147 | 0.048946 | 0.0383524 | 0.0635376 | 1908.61 | 0.1909 |
| MPREM | 0.7562075 | 0.8227116 | 0.3120231 | 0.008601 | 0.7581011 | 0.1555893 | 1.3817030 | 984.99 | 0.0985 |
| CGIOVERM | −0.3216062 | −0.0977224 | 1.4695690 | 0.054996 | −0.2946360 | −3.0623570 | 2.6019750 | 669.32 | 0.0669 |
| C | −0.0022033 | | 0.0230130 | 0.000776 | −0.0022647 | −0.0478770 | 0.0429293 | 952.04 | 0.0952 |
| Variance (Sigma2) | 0.0499917 | | 0.0064992 | 0.000139 | 0.0496235 | 0.0387848 | 0.0640200 | 2028.89 | 0.2029 |

Source: STATA output (author's compilation).

To assess how eligible the resulting posterior distributions are, we can analyze the convergence graphs of the models (Figures 1–4). Trace, autocorrelation and density plots are helpful to carrying out such a convergence diagnosis. Without any exceptions, all of the trace plots are homogeneous (dense vertical lines), thereby portray no evident trends or sparseness concluding well-mixing parameters. Favorably, autocorrelations become very small after about 15 lags and reach zero after a defined number of lags. Moreover, the density plots all indicate reasonably sufficient convergence for every model parameter because the density for the first half and the density for the second half seem identical.

As another empirical finding that supports good convergence, Table 9 depicts the results of credible interval hypothesis tests which provide robust evidence supporting the appropriateness of the posterior distributions proposed by the static models. All the probabilities computed for the parameters regarding their proposed lower and upper limits are very close to 95%, approving distribution appropriateness.

**Table 9.** Credible interval hypothesis tests.

| BIST 30 | 95% Credible Interval | | | TEST Statistics | |
|---|---|---|---|---|---|
| | Lower | Upper | Probability | Standard Deviation | Standard Error |
| MPREM | 0.2973357 | 1.6986190 | 0.9509 | 0.21609 | 0.0060980 |
| CGIOVERM | −3.2247780 | 3.3260440 | 0.9513 | 0.21525 | 0.0063197 |
| C | −0.0519224 | 0.0517887 | 0.9491 | 0.21980 | 0.0048018 |
| Sigma2 | 0.0513910 | 0.0834123 | 0.9478 | 0.22244 | 0.0041398 |
| **BIST Financials** | **95% Credible Interval** | | | **TEST Statistics** | |
| | Lower | Upper | Probability | Standard Deviation | Standard Error |
| MPREM | 0.4674241 | 1.6654280 | 0.9500 | 0.21796 | 0.0058677 |
| CGIOVERM | −3.2738730 | 2.5253270 | 0.9496 | 0.21878 | 0.0054853 |
| C | −0.0513845 | 0.0392210 | 0.9500 | 0.21796 | 0.0041725 |
| Sigma2 | 0.0385422 | 0.0633937 | 0.9497 | 0.21857 | 0.0044334 |
| **BIST Industrials** | **95% Credible Interval** | | | **TEST Statistics** | |
| | Lower | Upper | Probability | Standard Deviation | Standard Error |
| MPREM | 0.1935407 | 1.3790210 | 0.9499 | 0.21816 | 0.0051419 |
| CGIOVERM | −1.8068540 | 3.9258310 | 0.9498 | 0.21837 | 0.0049948 |
| C | −0.0338013 | 0.0585056 | 0.9501 | 0.21775 | 0.0046912 |
| Sigma2 | 0.0383524 | 0.0635376 | 0.9500 | 0.21796 | 0.0047583 |
| **BIST Services** | **95% Credible Interval** | | | **TEST Statistics** | |
| | Lower | Upper | Probability | Standard Deviation | Standard Error |
| MPREM | 0.1555893 | 1.3817030 | 0.9502 | 0.21754 | 0.0050913 |
| CGIOVERM | −3.0623570 | 2.6019750 | 0.9501 | 0.21775 | 0.0053417 |
| C | −0.0478770 | 0.0429293 | 0.9503 | 0.21734 | 0.0044676 |
| Sigma2 | 0.0387848 | 0.0640200 | 0.9500 | 0.21796 | 0.0444840 |

Source: STATA output (author's compilation).

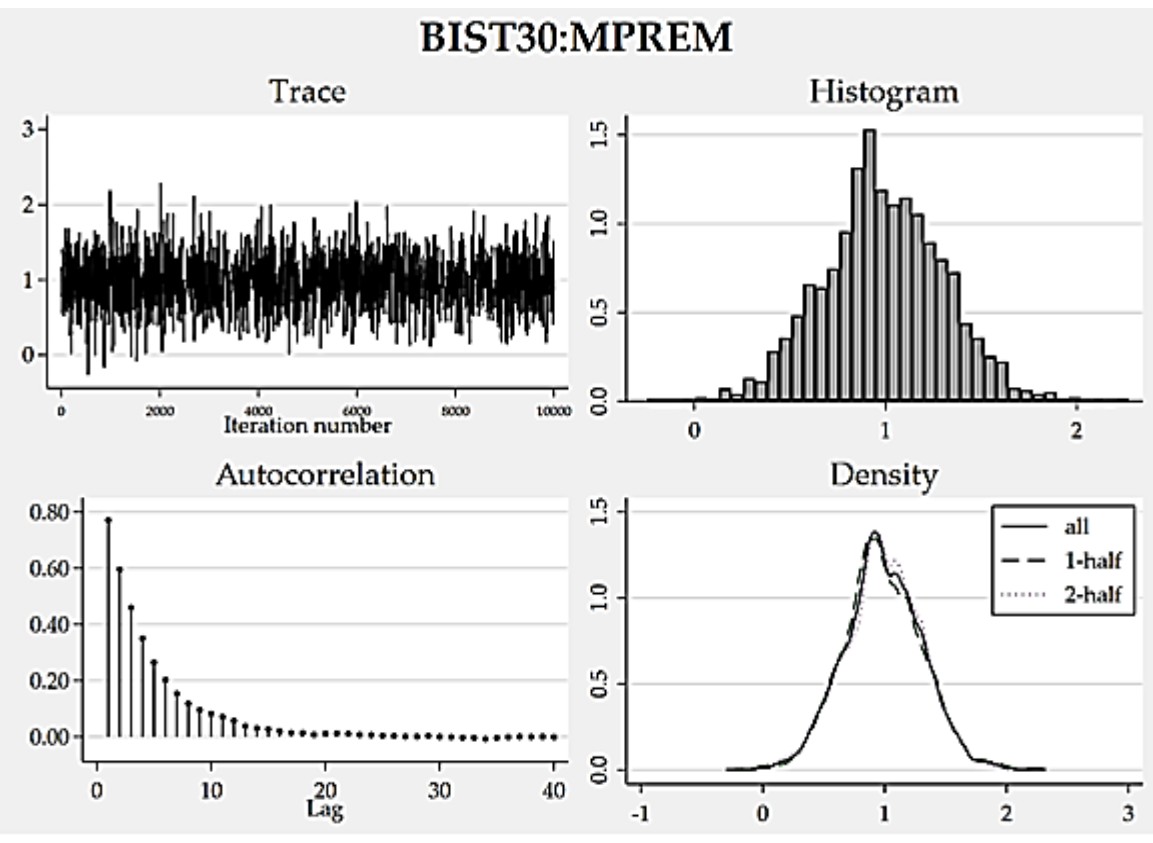

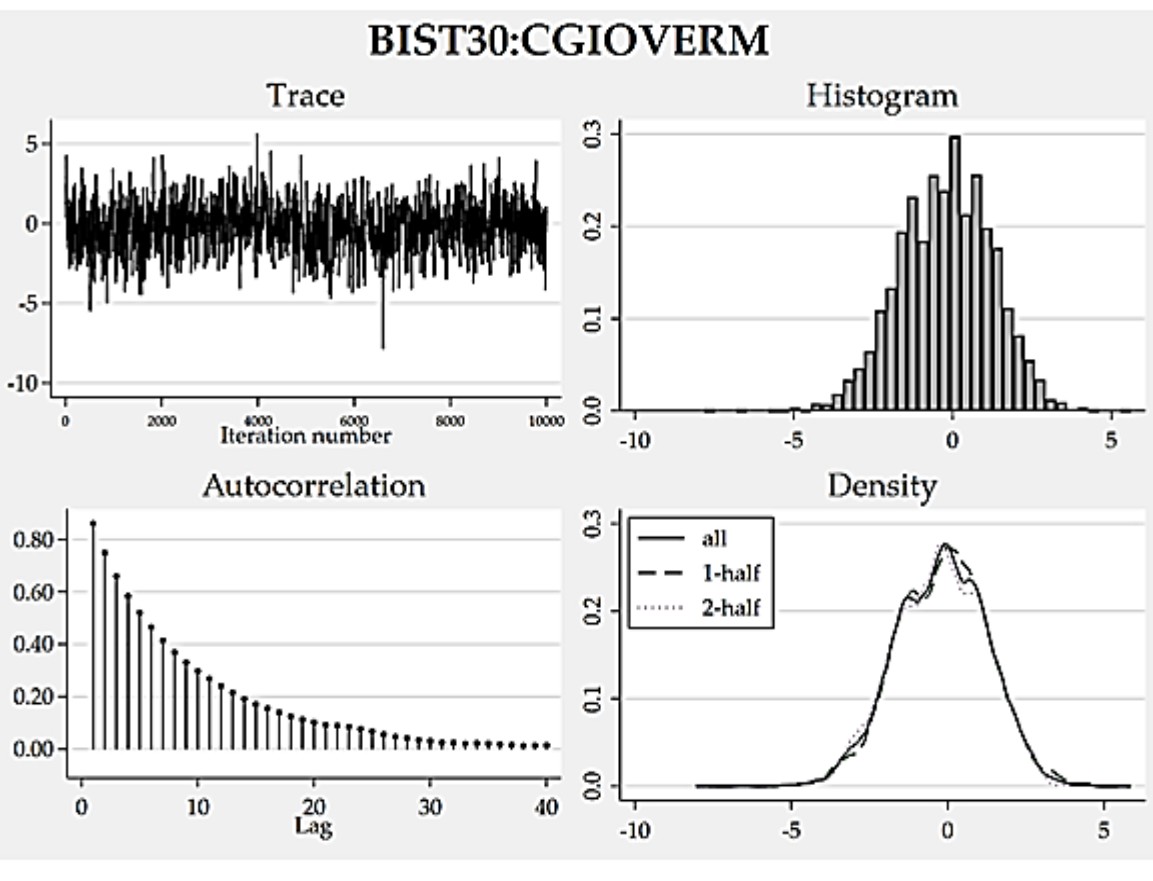

(**a**)

**Figure 1.** *Cont.*

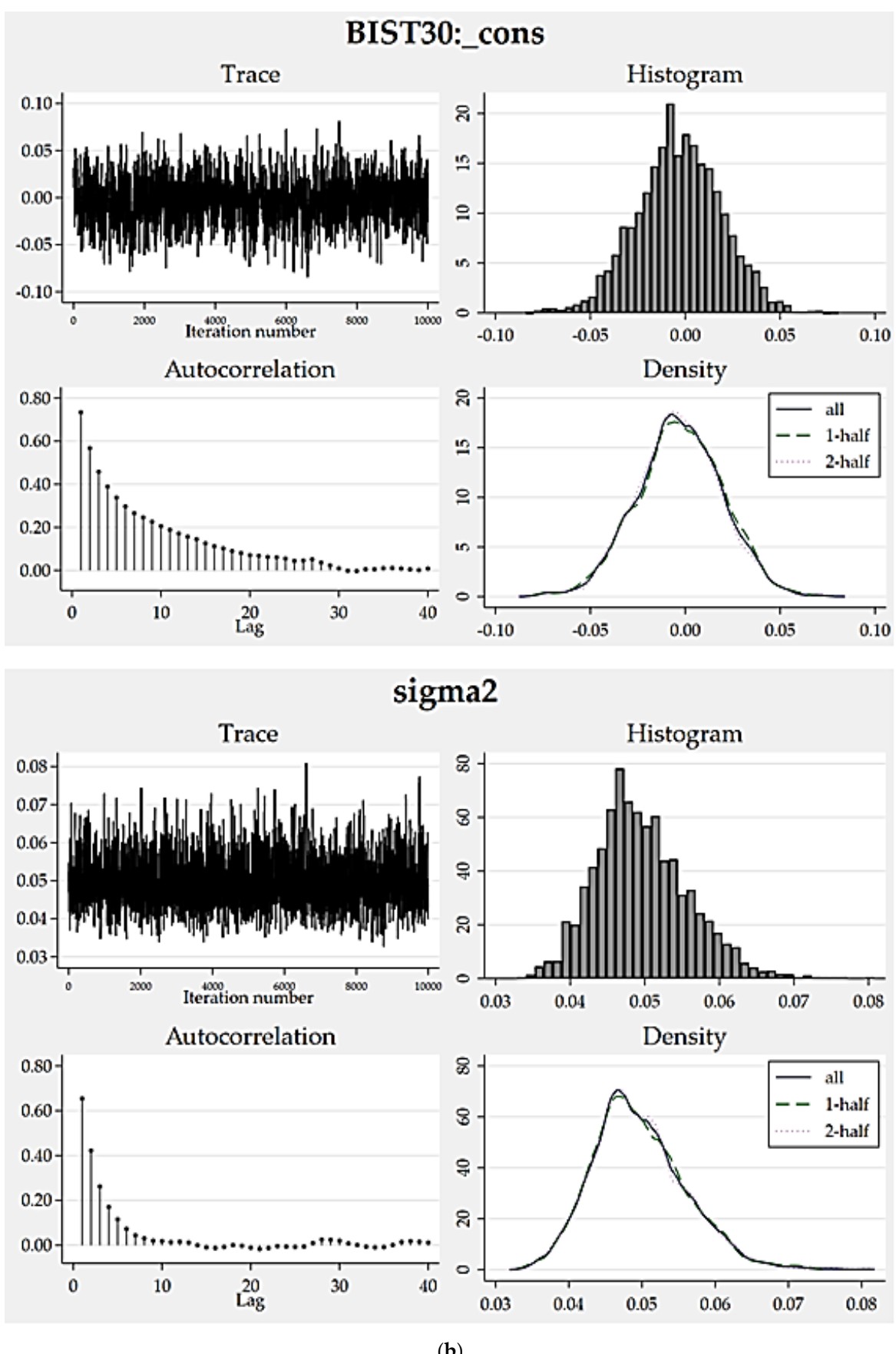

**Figure 1.** (**a**) BIST 30 convergence graphs for MPREM and CGIOVERM. (**b**) BIST 30 convergence graphs for model constant and variance. Source: STATA output (author's compilation).

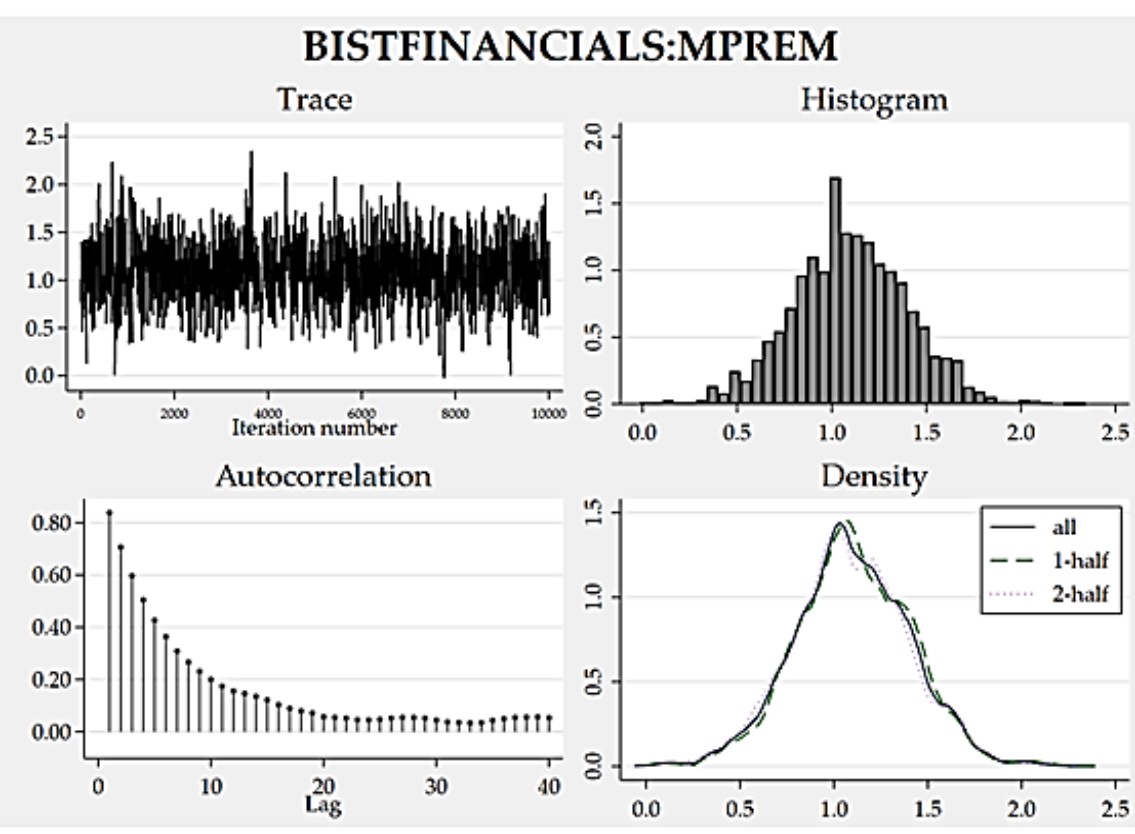

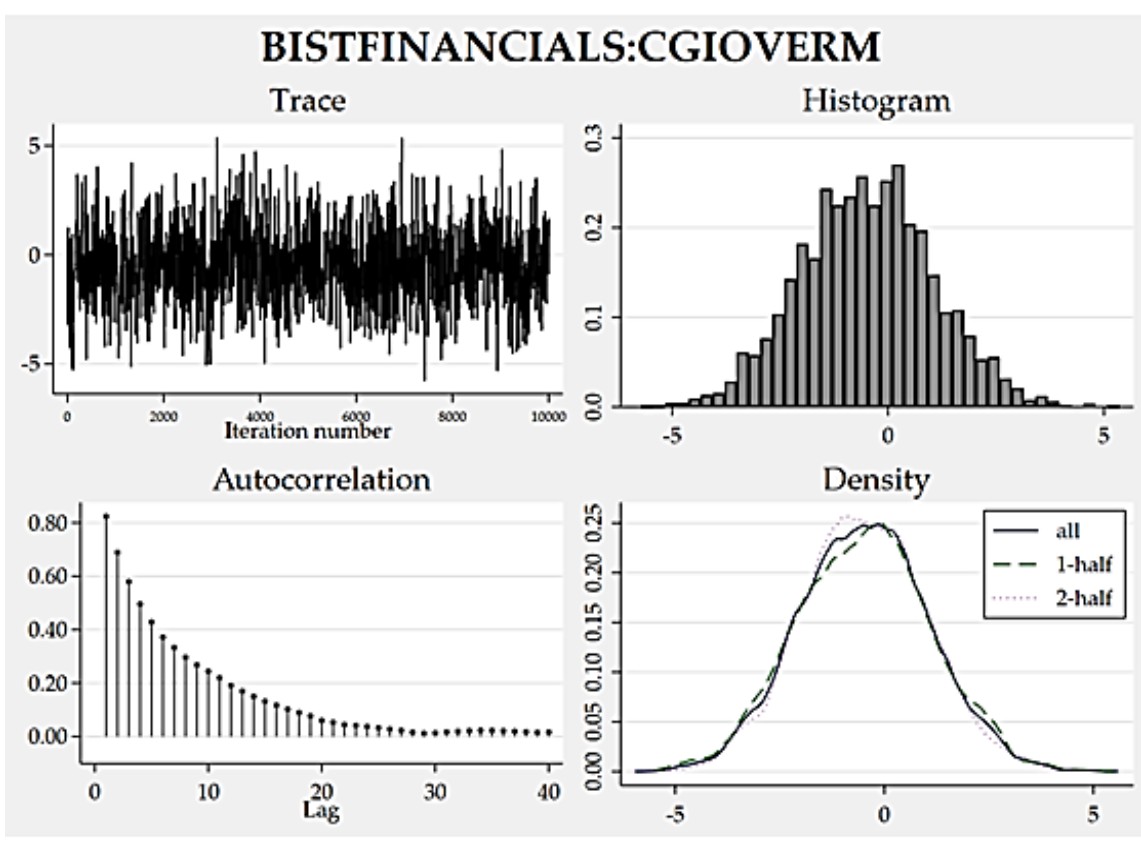

(**a**)

**Figure 2.** *Cont.*

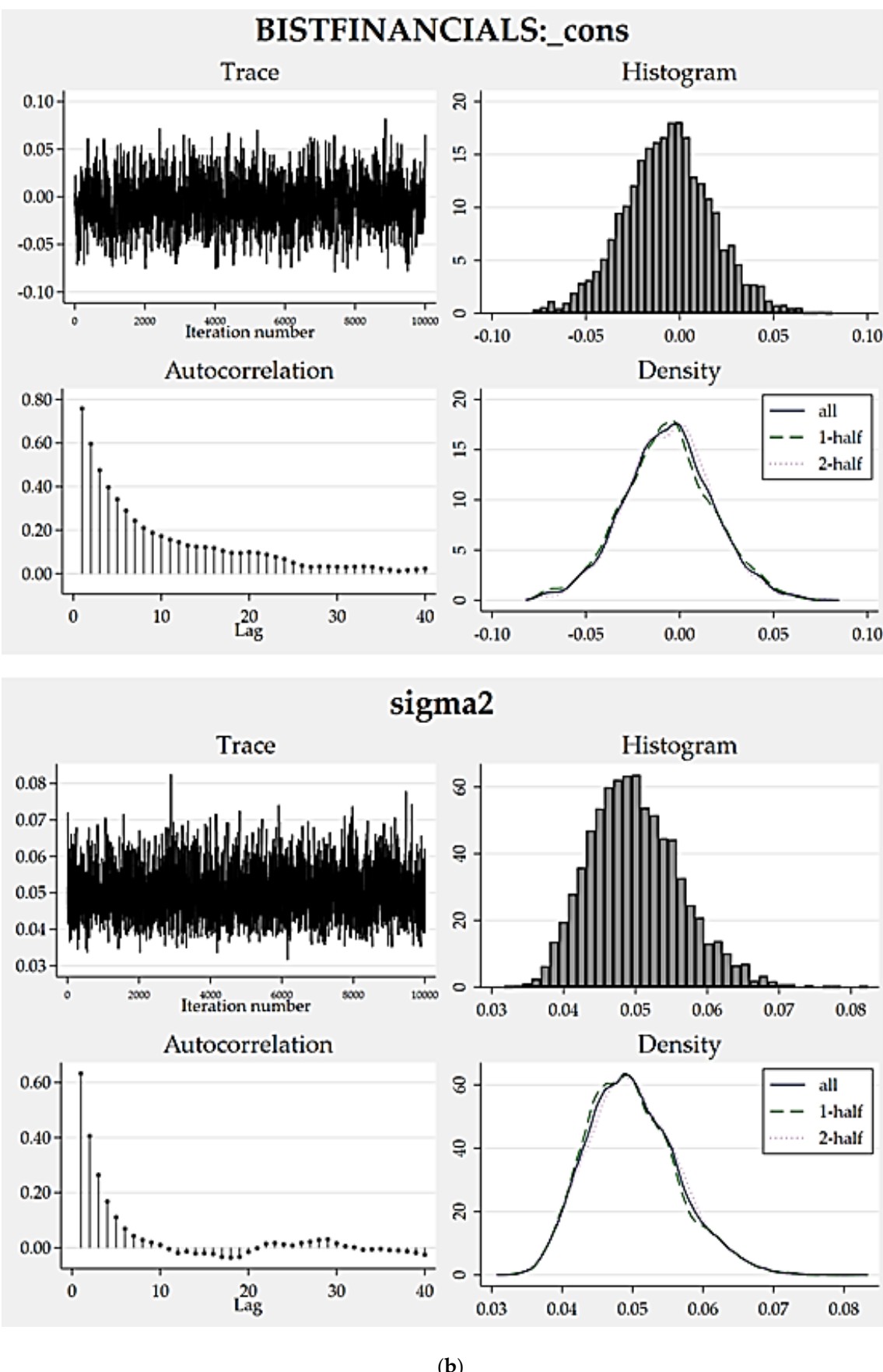

**Figure 2.** (**a**) BIST Financials convergence graphs for MPREM and CGIOVERM. (**b**) BIST Financials convergence graphs for model constant and variance. Source: STATA output (author's compilation).

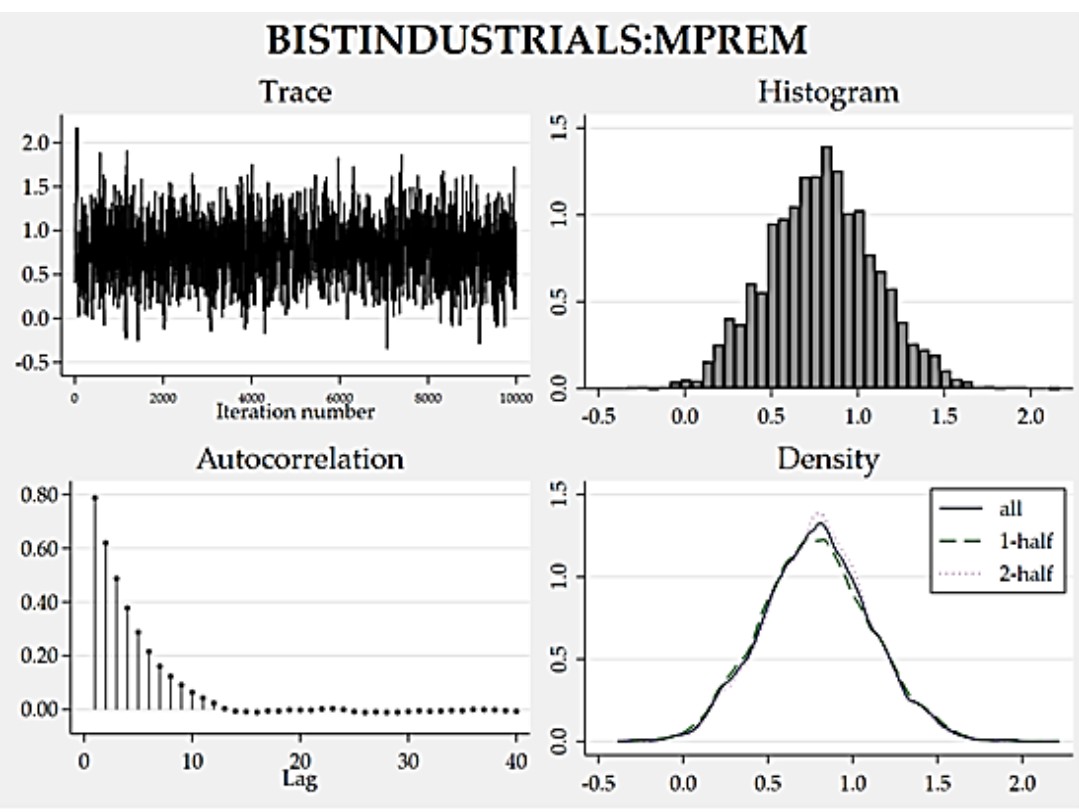

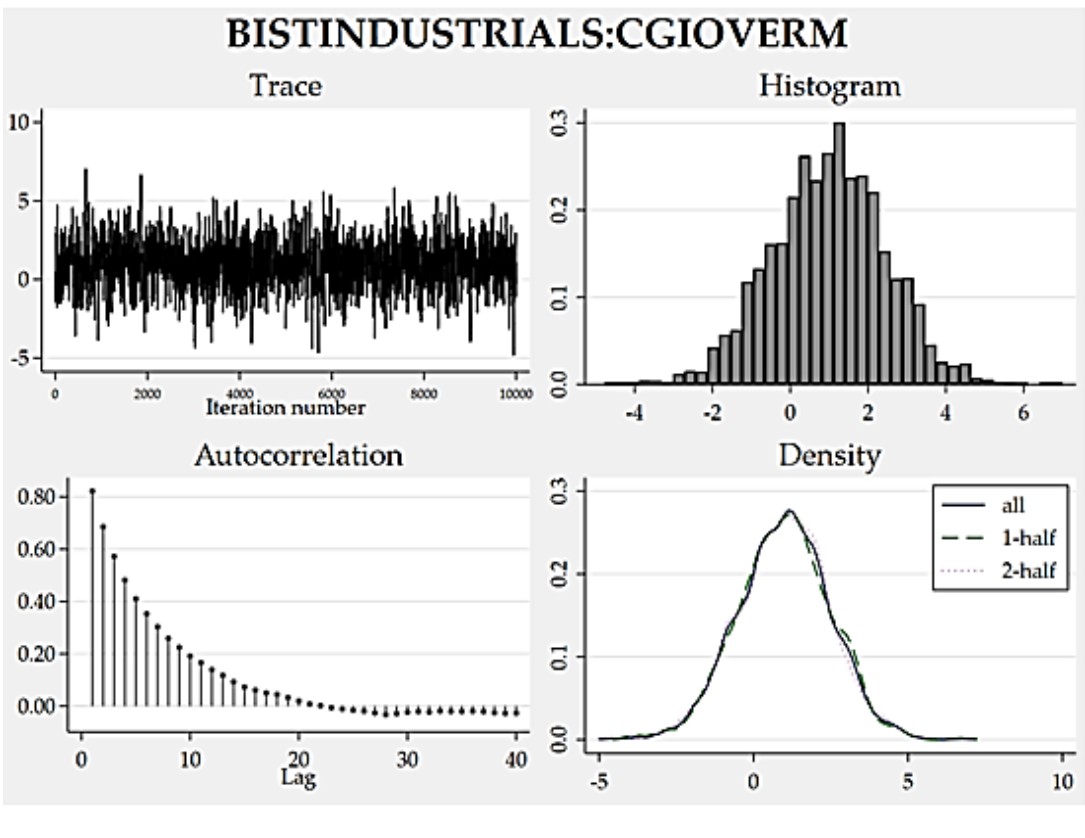

(**a**)

**Figure 3.** *Cont.*

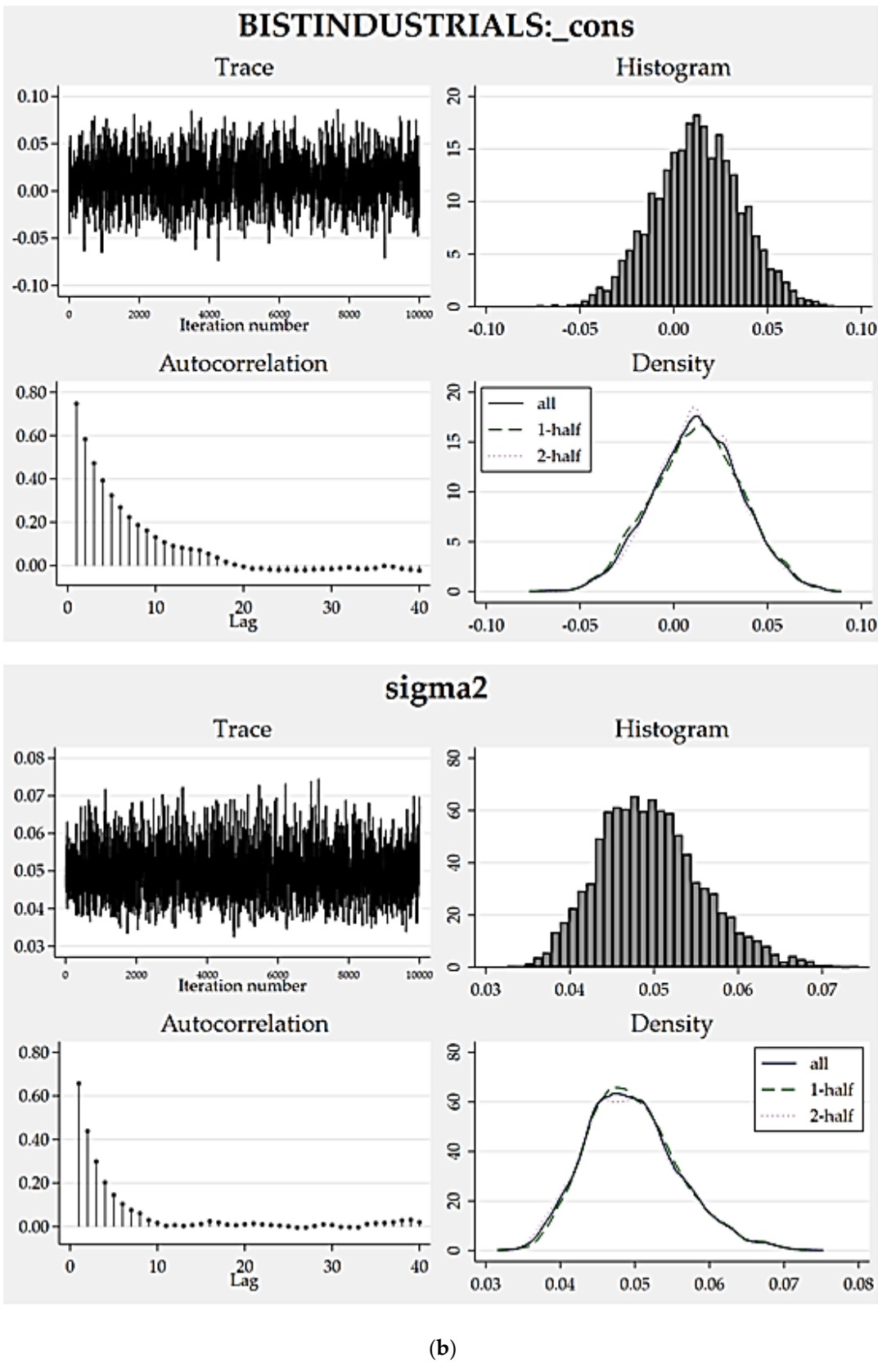

**Figure 3.** (**a**) BIST Industrials convergence graphs for MPREM and CGIOVERM. (**b**) BIST Industrials convergence graphs for model constant and variance. Source: STATA output (author's compilation).

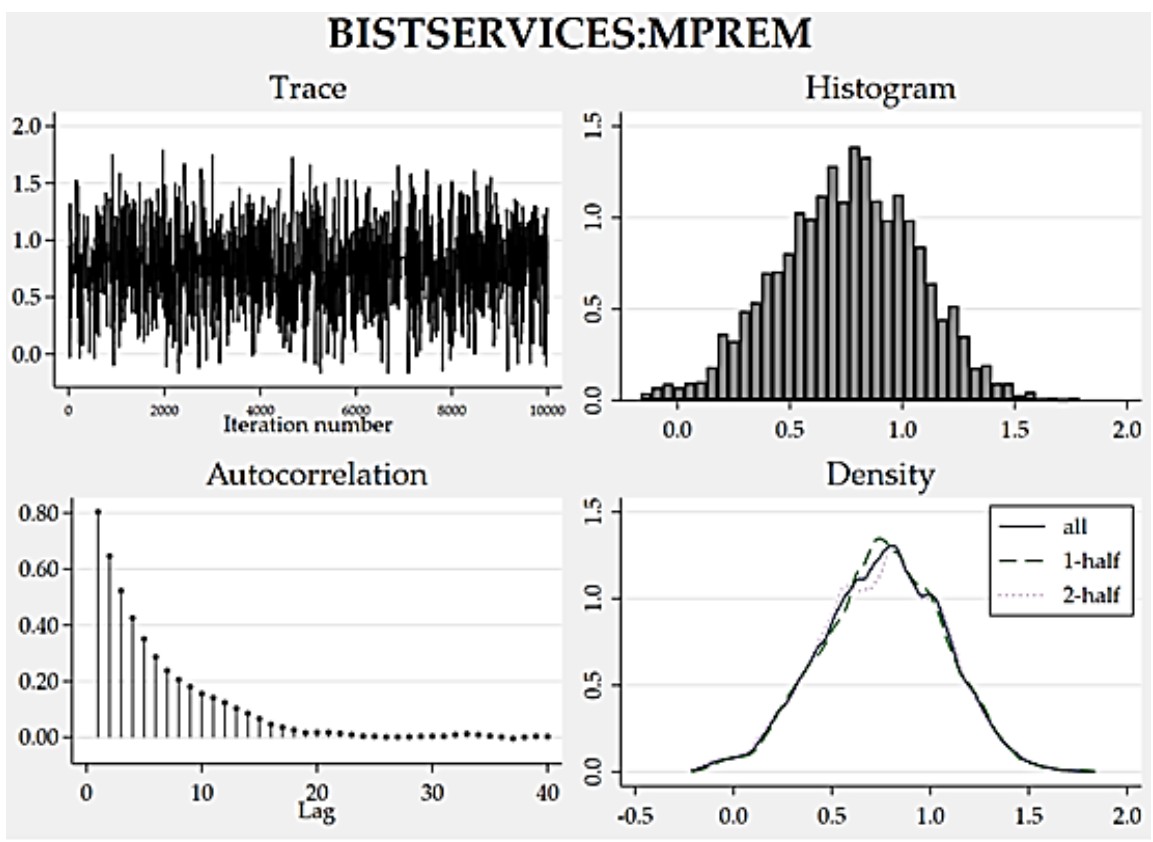

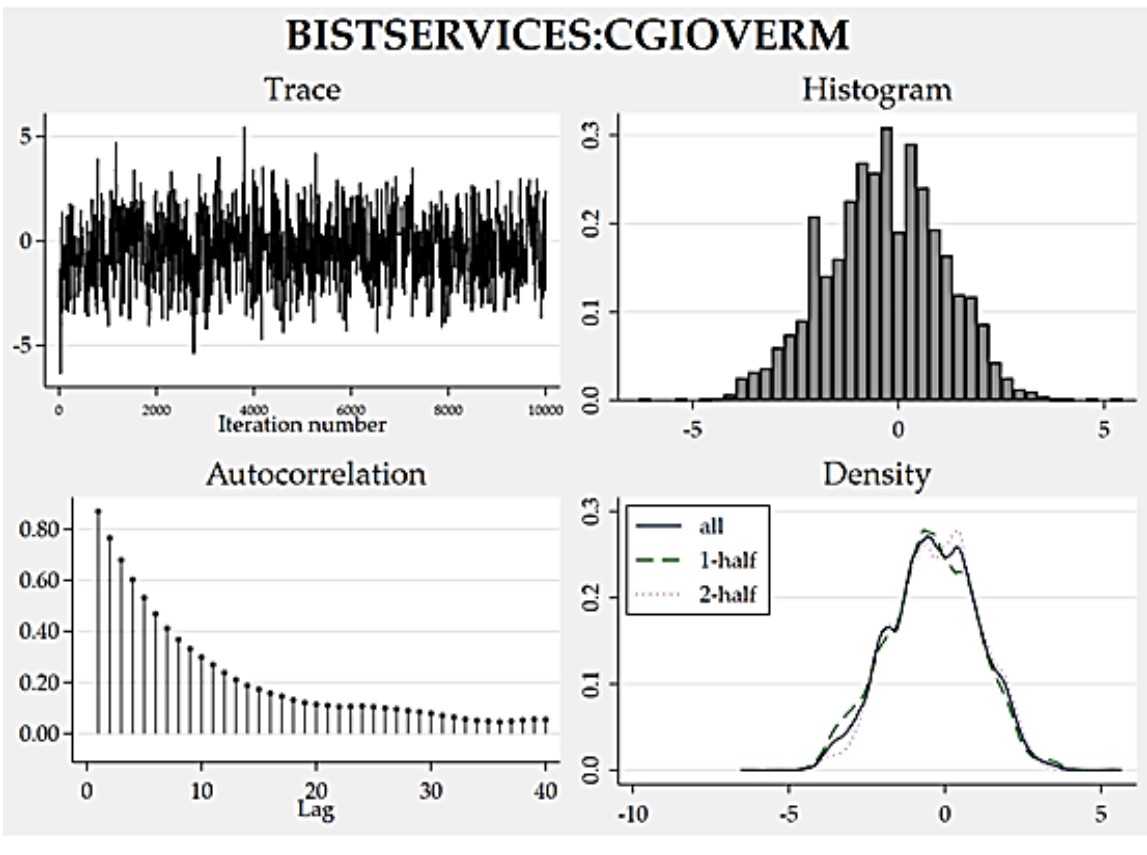

(**a**)

**Figure 4.** *Cont.*

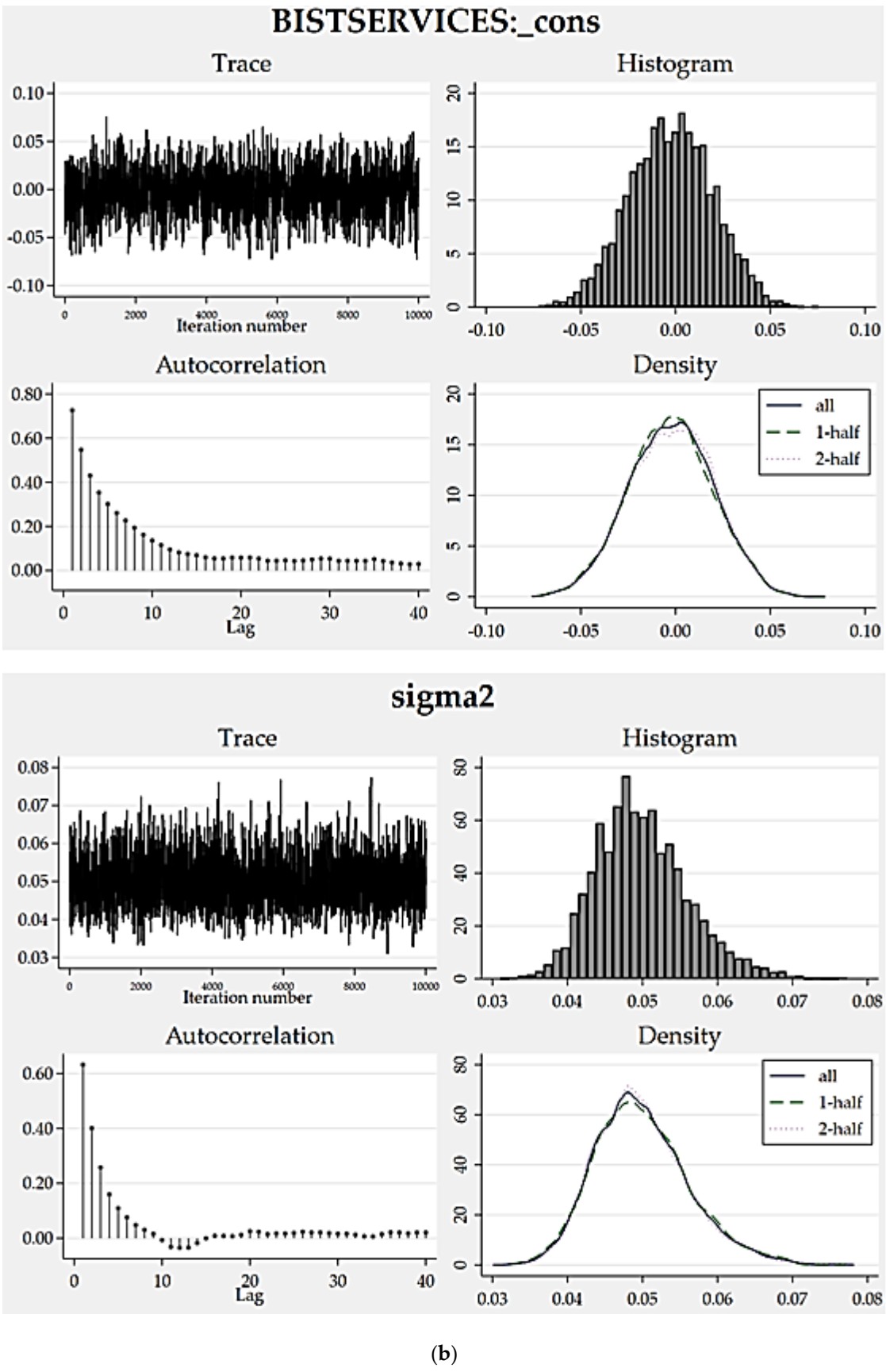

(b)

**Figure 4.** (**a**) BIST Services convergence graphs for MPREM and CGIOVERM. (**b**) BIST Services convergence graphs for model constant and variance. Source: STATA output (author's compilation).

### 5.2.3. Comparison between Robust and Bayesian Models

Taking into account the comparison statistics tabulated in Table 10, there exists no significant evidence for any remarkable difference between our robust and Bayesian model proposals in terms of their prediction performance and accuracy. For the BIST 30 index portfolio, the static Bayesian model achieves a better performance level as compared to its robust counterpart. Nonetheless, for the portfolios other than the BIST 30, the robust models slightly outperform the Bayesian model proposals. Furthermore, the coefficients of determination calculated for the BIST30, BIST Financials, and BIST Industrials portfolios are relatively high, but lower for the BIST Services portfolio. The Bayesian models could not perform better than the robust models, which means that no empirical proofs support the dominance of our Bayesian models over the robust models. This surprising conclusion may be considered ordinary and rational in view of our previous goodness-of-fit test results, suggesting approximation to normal distribution for all the variables. In other words, in cases where variable distributions are tested and proved to be approximately normal, Bayesian models may be a futile and tedious effort to achieve a performance level is substantially higher than those of classical econometric models.

**Table 10.** Comparison between robust and Bayesian models.

| Dependent Variable | Criterion * Model | RSS | MSE | Theil's R-square | Amemiya's PC | AIC | Adjusted R-square/Rw-square |
|---|---|---|---|---|---|---|---|
| BIST30 | ROBUST WITH VARYING SLOPE | 0.008508 | 0.000071 | 0.000071 | 0.008934 | 0.009080 | 0.9897 |
| | BAYESIAN STATIC | 0.004681 | 0.000037 | 0.000037 | 0.004836 | 0.004915 | 0.9912 |
| BIST Financials | ROBUST WITH VARYING SLOPE | 0.034083 | 0.000286 | 0.000284 | 0.035787 | 0.036375 | 0.9609 |
| | BAYESIAN STATIC | 0.037961 | 0.000319 | 0.000314 | 0.039216 | 0.039860 | 0.9427 |
| BIST Industrials | ROBUST WITH VARYING SLOPE | 0.040276 | 0.000338 | 0.000336 | 0.042290 | 0.042985 | 0.9138 |
| | BAYESIAN STATIC | 0.042350 | 0.000356 | 0.000350 | 0.043750 | 0.044468 | 0.8924 |
| BIST Services | ROBUST WITH VARYING SLOPE | 0.084781 | 0.000712 | 0.000707 | 0.089020 | 0.090482 | 0.8089 |
| | BAYESIAN STATIC | 0.088476 | 0.000743 | 0.000731 | 0.091401 | 0.092902 | 0.7786 |

Source: STATA output (author's compilation). * The lower the residual sum of squares (RSS). mean squared errors (MSE), Theil's R2, Amemiya's PC, and AIC values are. the better the model performance is. Theil's $R^2 = \frac{RSS}{n-k}$ Amemiya PC $= RSS\left(\frac{n+k}{n-k}\right)$ Akaike AIC $= RSS\left(e^{\frac{2(k+1)}{n}}\right)$ RSS $= \sum(y - \hat{y})^2$ MSE $= \frac{\sum(y-\hat{y})^2}{n-k-1}$. $k$: number of predictors. $n$: sample size.

### 5.3. An Illustration of How to Calculate and Interpret the Governance Quality Scores

Table 11 represents the governance quality scores computed for the portfolios of BIST 30, BIST Financials, BIST Industrials, and BIST Services by using the function depicted in Equation (11). As known, the scores are expected to change within the interval between 0 and 1. Any value approaching 1 means improved quality while decreases in the score should be interpreted as decayed governance quality. The larger the score is, the better quality the portfolio has. With respect to the calculated scores, BIST Industrials proves to be the portfolio with the highest degree of quality, but BIST Services has the lowest.

**Table 11.** Implied governance quality scores.

| Model | Robust Model | | | Bayesian Model | |
|---|---|---|---|---|---|
| Variable: CGIOVERM | Coefficient | *p*-Value | GQS | Standardized Coefficient | GQS |
| BIST 30 | −0.1290 | 0.0175 | 0.4684 | −0.0530 | 0.4867 |
| BIST Financial | −0.4595 | 0.0001 | 0.3871 | −0.0651 | 0.4837 |
| BIST Industrials | 0.9579 | 0.0000 | 0.7227 | 0.2333 | 0.5581 |
| BIST Services | −0.4181 | 0.0321 | 0.4001 | −0.0977 | 0.4756 |

Source: author's compilation.

## 6. Results and Discussion

Based on our research results, all the variable distributions are observed to approximate a normal distribution since the univariate Jarque–Bera and multivariate Doornik–Hansen test statistics are not significant at the 5% significance level. Also, all of the variable distributions seem to be almost symmetric. These results can be viewed as favorable for robust regression practices.

The univariate ADF unit root test results point out robust evidence sufficient to conclude stationarity for the variable distributions. Taking into account their significant ADF statistics, all the variables are observed to have no unit roots at levels in every case covering a constant and/or a time trend, which proves that they have no problem in working with time series to model relationships in linear form. We are also assured that there is no multicollinearity since the VIF scores are very low and much below 5.

Furthermore, in line with their Rn-square statistic, our robust model proposals are all detected to be accurate at the 1% significance level. However, for all of the portfolios, the conditional models with varying slopes outperform other version models if we consider their corresponding Rw-square statistics as well as significance of the predictors. Based on this result, it is more convenient to adjust the beta coefficient parallel to the country's foreign trade competitiveness position.

The coefficient estimates in all conditional robust models prove to be significant at 5%. The model constants can be stated as significant in the models, except the model for BIST Services, BIST 30, and BIST Financials are two portfolios which generated an average rate of return below what the respective model predicted. On the opposite side, BIST Industrials yielded a return, on average, above that predicted by the model. BIST Services can be considered to have generated no excess return over the predicted. BIST 30 and BIST Financials constitute two index portfolios whose sensitivities to market risk premium were predicted to climb for the sub-period with degraded competitiveness, but BIST Industrials and BIST Services deviated from the former two and attracted attention as having lower beta coefficients for the degradation period. Returns on BIST 30, BIST Financials and BIST Services are negatively affected by excess returns on BIST Governance over the market while the BIST Industrials returns are in positive relationship with them. The error terms are normally distributed at 5% for BIST Industrials and BIST Services, however we cannot conclude the same for BIST 30 and BIST Financials. The highest and lowest coefficients of determination are for BIST 30 and the BIST Services, respectively.

Having considered the acceptance and efficiency rates computed for our Bayesian model proposals as per model version and design, we can infer superiority of the model design with multivariate normal distribution assumption and blocking option over the others in all of the model versions, from their relatively high acceptance and efficiency scores. Plus, the static model version has higher BF, ML, and P(M/y) values, but lower DIC scores for all of the portfolios and hence surpasses the other versions. The Bayesian results signal no need for a conditional approach in computing the beta coefficient. Thus, the coefficient is estimated to remain constant and reveal no significant change during the whole sample period.

Parallel to the research results, the MPREM coefficient mean estimates indicate that all the portfolios except BIST Financials have a systematic risk level under that of the market portfolio. Since the credible intervals determined for the MPREM coefficient estimates do not include 0, this variable can be evaluated as statistically significant. Furthermore, the CGIOVERM coefficient estimates point out an inverse influential relationship among excess returns on BIST Governance over the market and returns on the BIST 30, BIST Financials, and BIST Services index portfolios. On the opposite side, the mentioned relationship is detected as positive for the BIST Industrials portfolio. However, there exist question marks concerning the insignificance of this predictor as its relative credible intervals involve 0 in every case. The same can be said for the model constant, which presents some evidence to conclude that none of the portfolios could generate a return, on average, over that predicted by the respective model. Since the efficiency rates computed for the parameters are approximately 7% and over, all the static model proposals can be suggested to be reasonably efficient.

Another result of the research is that all of the trace plots are homogeneous (dense vertical lines) thereby portray no evident trends or sparseness concluding well-mixing parameters. Favorably, autocorrelations get very small after about 15 lags and reach zero for a defined number of lags. Besides which, the density plots all emphasize a reasonably sufficient convergence for every model parameter as the density for the first half and the density for the second half seem quite identical.

As an evidence for good convergence, our research results concerning credible interval hypothesis tests are important. They present robust evidence supporting appropriateness of the posterior distributions proposed by the static models. All the probabilities calculated for the parameters regarding their proposed lower and upper limits are very close to 95%, approving distribution appropriateness.

To sum up in light of the above statistical diagnosis, according to the robust regression results, evident superiority of the conditional models over the static ones shows that the CAPM market betas may significantly differ with changes in the country's international trade competitive advantage. Furthermore, in both static and conditional robust model proposals the governance quality variable is found to be statistically significant, which demonstrates an outstanding relationship between portfolio returns and governance quality.

Our Bayesian findings favoring the static model version explicitly contradict the robust regression findings and thereby suggest stationary model constants and market betas, but insignificance of governance quality to stock returns. In other words, conditioned market betas could make no clear difference in model accuracy and performance.

Finally, there exists no significant evidence for any remarkable difference between our robust and Bayesian model proposals in terms of their prediction performance and accuracy. Concerning the BIST 30 index portfolio, the static Bayesian model showed a better performance level as compared to its robust counterpart. Indeed, when analyzing the portfolios other than the BIST 30, the robust models could outperform slightly the Bayesian model proposals. Moreover, the coefficients of determination are computed for the BIST 30. BIST Financials and BIST Industrials portfolios are relatively high, but lower for the BIST Services portfolio. The Bayesian models could not perform better than the robust models, which means that no empirical proofs support the dominance of Bayesian models over the robust models. This result can be evaluated as ordinary and rational from the perspective of former goodness-of-fit test results offering approximation to normal distribution for all the variables. In other words, in cases where variable distributions are proved to be approximately normal, Bayesian models can be a futile and tedious effort to reach a performance level is substantially higher than those of classical econometric models.

## 7. Concluding Remarks and Recommendations

The literature of capital asset pricing is interwoven with an enormous number of theoretical and empirical studies concentrating on the approaches to predicting asset returns. In this context, the CAPM and APT constitute two fundamental theories that try to model stock returns as based on certain systematic risk factors. The CAPM as a single-factor asset-pricing model uses market risk premium as the only factor (predictor) effective on stock returns and systematic risk exposure. On the other hand, the APT is a multifactor pricing model employing a more flexible methodology, but with a lower number of restrictive assumptions as compared to the CAPM. It allows us to cover more than one factor to estimate the effect of systematic risks on returns. In the existent literature, there are some other studies proposing different methodologies and/or functional forms which can be regarded as hybrid versions of the CAPM and APT. Unfortunately, there is no precise and unquestionable model offered that would lead to the best in predicting stock returns.

This paper aims to develop static and dynamic multifactor models to predict portfolio returns in the Borsa Istanbul that integrate the traditional CAPM with an additional factor calibrated to represent the effect of corporate governance quality through the robust and Bayesian linear regression techniques. As a result, at least two predictors were included in every model, one of which is excess returns on the market portfolio over the risk-free rate and the other is excess returns on the governance portfolio including stocks of the firms with an overall corporate governance quality score of at least 7, over that of the market portfolio. In our static models, the beta coefficient is not allowed to change during the entire sample period. But in the conditional or dynamic model versions, model constant and the beta coefficient are allowed to vary as to whether there is any degradation or improvement in the country's foreign trade competitive advantage. We also introduce an unsophisticated measure (governance quality score) based on our regression findings to approximately ascertain the portfolio's governance quality.

To construct the models, we used the monthly returns on the index portfolios; namely BIST 30, BIST Financials, BIST Industrials, BIST Services, BIST Governance, and BIST All from December 2009 to December 2019. BIST All is taken as the market portfolio while the BIST Governance is considered to be the governance portfolio. For the given period, we derived several robust and Bayesian models for each.

Our conditional dynamic robust models prevailed over other robust model candidates supporting the opinion that the CAPM beta coefficient may vary with changes in the national foreign trade competitive advantage. They also provided some evidence that portfolio returns are significantly affected by excess returns on BIST Governance over the market. On the other hand, our Bayesian model proposals favor the use of a static beta coefficient for the entire time interval and yield no clear proof for any significant effect of corporate governance quality on the portfolio returns. Furthermore, none of the Bayesian model proposals could achieve a performance level drastically over those by the robust models. At this point, we are convinced that it may be needless to utilize the Bayesian approach in predicting stock returns so long as variable distributions are proved to be normal.

We recommend further research to be designated to create similar models for other index portfolios in the Borsa Istanbul with some amendments on model specifications and technical choices. The research can be extended with inclusion of additional variables helpful to coping with the anomalies attributed to the traditional CAPM models.

**Author Contributions:** E.M. and C.K. are the main contributors in conceptualization and data analysis. B.C. collaborate with them on data gathering and analysis. All authors have read and agreed to the published version of the manuscript.

**Funding:** This research received no external funding.

**Institutional Review Board Statement:** Not applicable.

**Informed Consent Statement:** Not applicable.

**Data Availability Statement:** Data available in a publicly accessible repository that does not issue DOIs. Publicly available datasets were analyzed in this study. This data can be found here: www.investing.com (accessed on 25 December 2020).

**Conflicts of Interest:** The authors declare no conflict of interest.

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
