# Peer review of "Role of International Trade Competitive Advantage and Corporate Governance Quality in Predicting Equity Returns: Static and Conditional Model Proposals for an Emerging Market"

_jrfm, doi:10.3390/jrfm14030125_

Round 1

Reviewer 1 Report

I read the paper with pleasure., and believe the paper deserves publication in JRFM.

A minor question I have, concerns the normality assumption and lack of volatility clustering in the residual terms (the epsilons) in all regression  models (5),(6),(7),(8). I would think that the time series of the sample residuals will be better modeled by GARCH(1,1) with Student's T, or even better, with asymmetric GARCH with Normal Inverse Gaussian distributed innovations. For the multivariate case, a non-Gaussian copula in the residuals could be used.

On Table 1 the p-values of the multivariate normal fit (please change the decimal ", "with ".") vary strangely form 0.05 to 0.516 which is probably due to the non-normality and iid assumption of the residuals in the regressions (5)-(8).

I would like the authors to comments on those issues.

Author Response

  • On Table 1, the decimals of p-values of the multivariate statistics are changed to . from ,
  • Since the purpose of study is not to model volatility, we didn’t need to undertake GARCH models. In deriving the models, we assume the equilibrium based theories hold and the expected value of error terms is equal to zero. This arguement is tested through normality tests on error terms.
  • In the classic econometric models, if variables are proved to be distributed normally by normality tests, it is not necessary to worry about variable distributions. For our data, especially univariate distribution test results suggest normality.
  • We also made several revisions and improvements on the manuscript.

Reviewer 2 Report

The manuscript is poorly written and it is very hard to understand what the authors did in the paper which is novel/original. This paper may benefit substantially with the help of someone who has sufficient command on the English language. Their empirical tests are about finding correlations. Of course, such correlations are not necessarily causations. In other words, they need a convincing argument that their variable choices are interesting and add value to the literature. Otherwise, it merely becomes a fishing expedition. 

Author Response

(MINOR REVISIONS RECOMMENDED BY THE EDITOR)

  • On Table 1, the decimals of p-values of the multivariate statistics are changed to . from ,
  • Since the purpose of study is not to model volatility, we didn’t need to undertake GARCH models. In deriving the models, we assume the equilibrium based theories hold and the expected value of error terms is equal to zero. This arguement is tested through normality tests on error terms.
  • In the classic econometric models, if variables are proved to be distributed normally by normality tests, it is not necessary to worry about variable distributions. For our data, especially univariate distribution test results suggest normality.
  • In equation 1, the + sign is changed to – sign.
  • The numbering of equations is adjusted from equation 5 on.
  • The hypothesis section is removed and the hypothesis are integrated to study as arguments in the text.
  • We sepearated technical explanations about the methods employed and the research methodology into two distinct sections. We composed a new section named Methodology and Findings.
  • Methodology and Findings is presented in a stand-alone chapter.
  • Sources are added for all tables. The font in tables is also corrected and adjusted. A standard font is determined for all tables.
  • Graphs on Pages 17 to 20 are added to the appendix of the article. Graphs are also labeled and re-named as Figures.
  • To better highlight the results of research, a separate section named Results is added to the manuscript. So, the results of research are highlighted better and it is shown why the results are satisfactory.
  • Additional and up-to-date resources are included to the Literature Review section and the manuscript. Many scientific sources from the 21st century on this problem are added to the paper, which also comment on corporate governance.
  • We also made several revisions and improvements on the manuscript.

Reviewer 3 Report

Summary and overall opinion

The manuscript deals with developing a framework that combines a standard CAPM approach with corporate governance elements. The theoretical construction is fair and the research question might be interesting, though a bit subtle content-wise. The discussion is generally coherent but it also tends to be vague at times. The manuscript has a certain dose of intrinsic merit but is not publishable as it stands.

Comments and Suggestions 

2.1. Main concerns

  1. The research question in my opinion is rather subtle and should be heavily discussed.
  2. The introduction is one of the poorest sections of the manuscript. The main aspects that need to be addressed here are the following:
    1. The authors are not very successful in detecting a clear gap in the literature that the manuscript aims to fill.
    2. The discussion in terms of background is very thing.
    3. Most importantly, the authors fail to discuss the original contributions of the manuscript and their relevance in relation to the current state of knowledge in the field.
    4. The section contains elements that are misplaced and would be more useful in the Framework section.
  3. The general composition and labeling of sections are very poor, misleading, and confusing and the manuscript exhibits a very consistent need for upgrades in drafting. I advise the authors to try and form a stable, clear-cut, and solid Data and Methodology section and separate it from literature, theoretical considerations, or other elements that do not belong in such a section.
  4. I want to see a self-standing "Results" section, that as the name goes contains the results deriving from the methodology.
  5. The current results section is very poor and full of econometric output and an excess of redundant, basic econometric talk. I want to see here a clear presentation of the actual results of the paper. The authors should focus less and basic econometric elements with little added value and allocate more room to actual results. 
  6. The manuscript needs a rigorous Discussion section. 
  7. The conclusion section also requires upgrades. I would like to see clear takeaways that derive from the study
  8. The manuscript would benefit from (heavy) robustness testing.

2.2. Minor concerns

1. The manuscript contains elements that can be easily considered general knowledge or redundant. They make the manuscript hard to follow at times while contributing little value. Elements such as:

"A typical Bayesian modeling process begins with specifying prior and posterior distributions for the parameters" (335) or

"Ordinary Least Squares (OLS) as one of the foremost econometric methods" (311) could be easily dropped. 

2.Tables 1 to 11 should be considered as appendices. 

3. Figures 1 to 4 should be considered as appendices. 

Author Response

(MINOR REVISIONS RECOMMENDED BY THE EDITOR)

  • On Table 1, the decimals of p-values of the multivariate statistics are changed to . from ,
  • Since the purpose of study is not to model volatility, we didn’t need to undertake GARCH models. In deriving the models, we assume the equilibrium based theories hold and the expected value of error terms is equal to zero. This arguement is tested through normality tests on error terms.
  • In the classic econometric models, if variables are proved to be distributed normally by normality tests, it is not necessary to worry about variable distributions. For our data, especially univariate distribution test results suggest normality.
  • In equation 1, the + sign is changed to – sign.
  • The numbering of equations is adjusted from equation 5 on.
  • The hypothesis section is removed and the hypothesis are integrated to study as arguments in the text.
  • We sepearated technical explanations about the methods employed and the research methodology into two distinct sections. We composed a new section named Methodology and Findings.
  • Methodology and Findings is presented in a stand-alone chapter.
  • Sources are added for all tables. The font in tables is also corrected and adjusted. A standard font is determined for all tables.
  • Graphs on Pages 17 to 20 are added to the appendix of the article. Graphs are also labeled and re-named as Figures.
  • To better highlight the results of research, a separate section named Results is added to the manuscript. So, the results of research are highlighted better and it is shown why the results are satisfactory.
  • Additional and up-to-date resources are included to the Literature Review section and the manuscript. Many scientific sources from the 21st century on this problem are added to the paper, which also comment on corporate governance.
  • The sections of the manuscript are re-organized and rearranged.
  • We also made several revisions and improvements on the manuscript.

Reviewer 4 Report

The presented article deals with an interesting topic. The application of CAPM and APT is one of the main topics often discussed - however, the solutions derived for this purpose are often difficult to apply in practice because they are based on unreal assumptions. Thus, the entire prediction of individual returns, as well as portfolio returns, can only be further explored under strict laboratory conditions.

I would have the following formal and substantive comments on the article:

- Page 2, equation 1 - a dangerous mistake - the risk premium is the difference between market return and risk-free return and not the sum of both!

- From the formal point of view, the numbering of the equations needs to be adjusted - from equation 5 on, they are badly formatted.

- In subsection 3.1, hypotheses are promised in the title, but they are nowhere to be found in the text-although they could be derived, as quantitative research allows.

- In subchapter 3.2, the research methodology is addressed. Nevertheless, in the text, section 3.3.3 is labeled Methodology. Why?

- The third chapter should be corrected completely and should focus on methodology. The results of the research should be presented in a stand-alone chapter.

- All tables lack information about source, whether it is own research. The font in the tables is a combination of Calibri, Times New Roman and Arial - adjust all tables to one format.

- Pages 17 to 20 contain graphs that should be added to the appendix of the article. For the explanation of the results, a commentary would be much more helpful than 16 different graphs.

- In the Conclusion, the authors should better highlight the results of their research so that it is clearer to the reader why the results are satisfactory. At the same time, the verification/falsification of the hypotheses should also take place here.

- The Inputs for further research are very brief with low information content.

- It is a pity that the authors used only ancient sources. From a total of 47 sources, 29 sources are older than 20 years! There are many scientific sources from the 21st century on this problem, which also comment on corporate governance!

Author Response

  • In equation 1, the + sign is changed to – sign.
  • The numbering of equations is adjusted from equation 5 on.
  • The hypothesis section is removed and the hypothesis are integrated to study as arguments in the text.
  • We sepearated technical explanations about the methods employed and the research methodology into two distinct sections. We composed a new section named Methodology and Findings.
  • Methodology and Findings is presented in a stand-alone chapter.
  • Sources are added for all tables. The font in tables is also corrected and adjusted. A standard font is determined for all tables.
  • Graphs on Pages 17 to 20 are added to the appendix of the article. Graphs are also labeled and re-named as Figures.
  • To better highlight the results of research, a separate section named Results is added to the manuscript. So, the results of research are highlighted better and it is shown why the results are satisfactory.
  • Additional and up-to-date resources are included to the Literature Review section and the manuscript. Many scientific sources from the 21st century on this problem are added to the paper, which also comment on corporate governance.
  • We also made several revisions and improvements on the manuscript.

Round 2

Reviewer 2 Report

I am satisfied the changes made. The paper makes a contribution consistent with the aims and scopes of the journal.

Author Response

REVISIONS MADE

  1. The abstract was revised. The number of words was reduced under 200.
  2. The research question has been emphasized in the introduction section.
  3. Some of the text in the ‘introduction’ section was moved to “Conceptual Framework” section.
  4. Tables were moved to the end of the paper as a separate appendix.
  5. Some paragraphs were added to the end of the Results section to sum up research results in finance perspective as discussion.
  6. Several redundant statements were dropped.
  7. Sevaral language and spelling revisions were performed.

Reviewer 3 Report

Summary and overall opinion

First of all, I would like to congratulate the authors for their efforts put into providing a more refined version of the manuscript. The authors made a good effort in tackling several of the concerns I expressed regarding the initial version of the paper. My present review will follow in the line of the comments put forward in my original review, trying to treat them in the same order. 

Comment 1. remains not answered. 

Comment 2. The authors brought several improvements to the original version. However, these are far from what was instructed. Given these circumstances, I consider this point to be partially treated.

Comment 3 was dealt with in the revision.

Comment 4 was dealt with in the revision.

Comment 5 is as best partially considered.

Comment 6 remains not answered. 

Comment 7 remains not answered.

Comment 8 remains not answered.

The minor concerns have been partially addressed.

Author Response

(The authors gave the same response as above.)
